# Global diversity of soil-transmitted helminths reveals population-biased genetic variation that impacts diagnostic targets

Soil-transmitted helminths (STHs) are intestinal parasites that affect over a billion people worldwide. STH control relies on microscopy-based diagnostics to monitor parasite prevalence and enable post-treatment surveillance; however, molecular diagnostics are rapidly being developed due to increased sensitivity, particularly in low-STH-prevalence settings. The genetic diversity of helminths and its potential impact on molecular diagnostics remain unclear. Using low-coverage genome sequencing, we assess the genetics of STHs within worm, faecal, and purified egg samples from 27 countries, identifying differences in the genetic connectivity and diversity of STH-positive samples across regions and cryptic diversity between closely related human- and pig-infective species. We define substantial copy number and sequence variants in current diagnostic target regions and validate the impact of genetic variation on qPCR diagnostics using in vitro assays. Our study provides insights into the diversity and genomic epidemiology of STHs, highlighting both the challenges and opportunities for developing molecular diagnostics needed to support STH control efforts.

Soil-transmitted helminths (STHs), which include the giant roundworm (*Ascaris lumbricoides*), threadworm *(Strongyloides stercoralis)*, whipworm (*Trichuris trichiura*), and hookworms (*Necator americanus*, *Ancylostoma duodenale* and *Ancylostoma ceylanicum*), collectively infect nearly a quarter of the world's human population. Infections occur through the consumption of infective eggs or the penetration of the skin by infective larvae, which then migrate to the host's intestines, where STHs sexually mature, reproduce, and lay eggs that are passed in the faeces[1–3]. *Strongyloides* species are unique in that their eggs hatch within the host, and larvae pass into the faeces, some of which can develop directly into infectious larvae and autoinfect the host[4]. Light-intensity STH infections are often asymptomatic, but moderate-to-heavy intensity infections can cause significant morbidity[5], including anaemia, malaise and weakness, intestinal obstruction, malnutrition, impaired growth and developmental delay[6]. Control of STHs relies almost exclusively on annual or biannual mass drug administration (MDA) of anthelmintics to at-risk populations (preschool and school-aged children (SAC) as well as women of reproductive age) or to entire communities to reduce morbidity and transmission[7], along with health education and the improvement of water, sanitation and hygiene conditions[8]. This control strategy is central to the World Health Organisation's (WHO) 2030 Roadmap for STHs, which aims to achieve and maintain the elimination of morbidity caused by STHs as a public health problem and to establish an efficient strongyloidiasis control programme in SAC by 2030[9].

A critical aspect of the WHO's control strategy for STHs is the use of diagnostics to detect and monitor infections, from establishing baseline prevalence to confirming elimination as a public health problem once MDA campaigns have concluded[10]. The WHO currently recommends microscopy-based diagnostic methods, which are particularly suited for diagnosing moderate-to-heavy intensity infection burdens due to their simplicity and relatively low cost. The Kato-Katz light microscopy technique is widely used to detect helminth eggs in faecal smears[11]. However, its sensitivity is significantly reduced when infection burdens are low[12], leading to variability in parasite detection. Furthermore, this approach is ineffective for diagnosing *Strongyloides*

✉ e-mail: mpapaiakovou@gmail.com; stephen.doyle@sanger.ac.uk

infections due to the intermittent excretion of larvae and low parasite load[13]. DNA-based diagnostics, such as quantitative polymerase chain reaction (qPCR) assays, are beginning to offer an alternative to conventional microscopy-based STH diagnosis methods due to the increase in their sensitivity and specificity in low prevalence settings[14] and their potential for application during post-deworming surveillance where microscopy approaches become less efficient owing to the reduced intensity of infection[15–19]. Current qPCR assays were primarily developed and validated using a single or limited number of geographically restricted parasite isolates[20–24]. However, it is increasingly recognised that parasitic worms of humans[25–30] and animals[31–33] are genetically diverse and distributed within genetically structured populations at local and global scales. These genetic variations can occur in sequences targeted by molecular methods, and therefore, the variation within a single species that differentiates STH populations may also affect the sensitivity and specificity of diagnostic tests in different settings. To date, there have been only a few studies on the population genetics of STHs[34], limiting our understanding of the extent of genetic variation that might interfere with molecular diagnostic methods.

To understand the population genetics of STHs and how genetic variation impacts molecular diagnostics, we analysed low-coverage whole-genome and metagenomic sequencing data of adult worm ($n = 128$), faecal ($n = 842$), and purified egg ($n = 30$) samples from 27 countries worldwide, of which almost all are endemic for STHs. Analysis of these data revealed the abundance and diversity of single- and mixed-species STH infections, the genetic connectivity of STH populations across different geographic scales, and significant genetic variation at sites targeted by diagnostics, both within genomes and amongst countries. Utilising in vitro assays, we evaluated the impact of genetic variants on qPCR diagnostics. These data will aid molecular epidemiological and emerging genomic efforts to elucidate STH transmission patterns and help validate current and future diagnostic tools to support the control and elimination of STHs.

## Results

### Detection of single- and co-infections of STHs from faecal, egg and worm isolates

To understand the genetic diversity and differentiation of STH populations worldwide, we analysed data from 1000 DNA samples with known or suspected STH infections. This sample cohort included 150 samples from 13 countries (including adult worm and faecal samples) selected for sequencing based on confirmed helminth infections via microscopy or qPCR, along with 850 publicly available genomic datasets (including purified eggs, adult worms, and faecal samples) from 19 countries. In total, the 1000 genomic datasets, derived from 128 individual adult worms, 842 faecal samples, and 30 semi-purified egg samples, represented 27 countries (Supplementary Data 1) from six continents.

Sequencing reads from each dataset were mapped to eight mitochondrial and nuclear reference genomes of STHs and other gastrointestinal helminths reported to infect humans (Supplementary Data 2). Out of 842 faecal samples, 175 were positive by sequencing for at least one helminth species (minimum threshold of 10 helminth reads / million reads mapped; Supplementary Fig. 1a) spanning 14 different countries (Fig. 1a, c and Supplementary Fig. 2a). We suspect that our approach resulted in false-negatives, due to (i) the low parasite prevalence of many infections and in turn, low proportion of parasite DNA present particularly in complex faecal samples, and (ii) the conservative minimum read threshold necessarily to ensure sensitivity and specificity of detection. Reads mapping to *A. lumbricoides* were the most abundant (96 single-infections and 27 co-infections containing two or more species), which might be expected given that *Ascaris* worms are highly fecund (females deposit >200,000 eggs per day)[35] and would result in a larger proportion of *Ascaris* DNA being

sequenced relative to other parasites present. The next largest number of infections was caused by *N. americanus* (35 single infections and 13 co-infections), followed by *T. trichiura* (six single infections and 15 co-infections). We also assessed whether we could detect other helminth species present in faecal samples by sequencing, mapping reads to *Schistosoma mansoni* (eight single infections and one co-infection), *Strongyloides stercoralis* (one co-infection), *Enterobius vermicularis* (two co-infections), and *Ancylostoma ceylanicum* (one single infection and two co-infections). Among the 158 samples derived from worms or pools of eggs, 154 were helminth-positive by sequencing, spanning 15 different countries (Fig. 1b, d and Supplementary Fig. 1b). *A. lumbricoides* was the most abundant helminth present in the cohort tested (71 single infections and 20 co-infections), followed by *T. trichiura* (26 single infections and 22 co-infections), *S. stercoralis* (16 single infections), *N. americanus* (five single infections, nine co-infections), *S. mansoni* (11 single infections, one co-infection), and one co-infection each, by *E. vermicularis* and *Ancylostoma duodenale*.

### Mitochondrial genetic diversity within and between populations of *Ascaris* spp. and *Trichuris trichiura*

To estimate the genetic diversity within and between populations of each STH species, we focused our analyses on samples with a minimum mean 5x sequencing coverage to support variant calling. From the sample cohort samples, we retained 108 *A. lumbricoides*-positive adult worm, concentrated egg and faecal samples from 10 countries, 35 *T. trichiura*-positive adult worm, egg and faecal samples from seven countries, 16 *S. stercoralis*-positive adult worm samples from two countries, 12 *S. mansoni*-positive adult worm and faecal samples from seven countries, and seven *N. americanus*-positive adult worm, concentrated egg and faecal samples from two countries. Focusing on *A. lumbricoides and T. trichiura*, we identified 2054 robust mitochondrial single nucleotide polymorphisms (SNPs) (Supplementary Data 3) for population genetic analyses: 558 SNPs in *A. lumbricoides* positive samples ($n = 88$ samples) and 1496 SNPs ($n = 30$ samples) in samples positive for *T. trichiura*. Considering that no SNPs from *N. americanus* remained after stringent filtering (Supplementary Data 3), *S. stercoralis*-positive samples represented only two countries (15 samples from Thailand and one from the USA), and the population genetic analysis of *S. mansoni* datasets has already been explored[36], these helminths were excluded from further analysis.

Initial analyses of *A. lumbricoides* genetic variation estimated by mapping to a Korean mitochondrial reference genome of this species isolated from a human (Accession: NC_016198[37]) showed unusually low between-country and high within-country (specifically, within Kenya) genetic diversity and a lack of country-specific population clustering (Supplementary Figs. 3a–c). These clustering patterns could not be attributed to missing variants or coverage biases (Supplementary Fig. 3d). *Ascaris* is a known zoonotic parasite, and there is ongoing debate regarding whether human-infective *A. lumbricoides* and the pig-associated *A. suum* represent a single or distinct species[38,39]. To explore potential reference mapping biases, we mapped *Ascaris*-positive samples to two additional *Ascaris* mitochondrial reference genomes, one isolated from a pig (*A. suum* from the US; Accession: NC_001327[40];) and one from a human (*A. lumbricoides* from Tanzania; Accession: KY045802). Competitive mapping to the three references showed a bias for the *A. suum* reference rather than either of the *A. lumbricoides* references (Supplementary Fig. 4a). Only six samples mapped preferentially to the Korean *A. lumbricoides* reference (Supplementary Fig. 4b). The remaining samples fit two *A. suum*-like profiles: (i) majority *A. suum* with partial Tanzania *A. lumbricoides*, and (ii) majority *A. suum* only. The first was primarily driven by the Kenyan samples, previously described as displaying *A. suum*-like variation[39]; however, we also observed the same pattern in approximately half the samples from Myanmar. The remaining countries fit the second profile, with minimal *A. lumbricoides* variation present. The six Kenyan outliers that

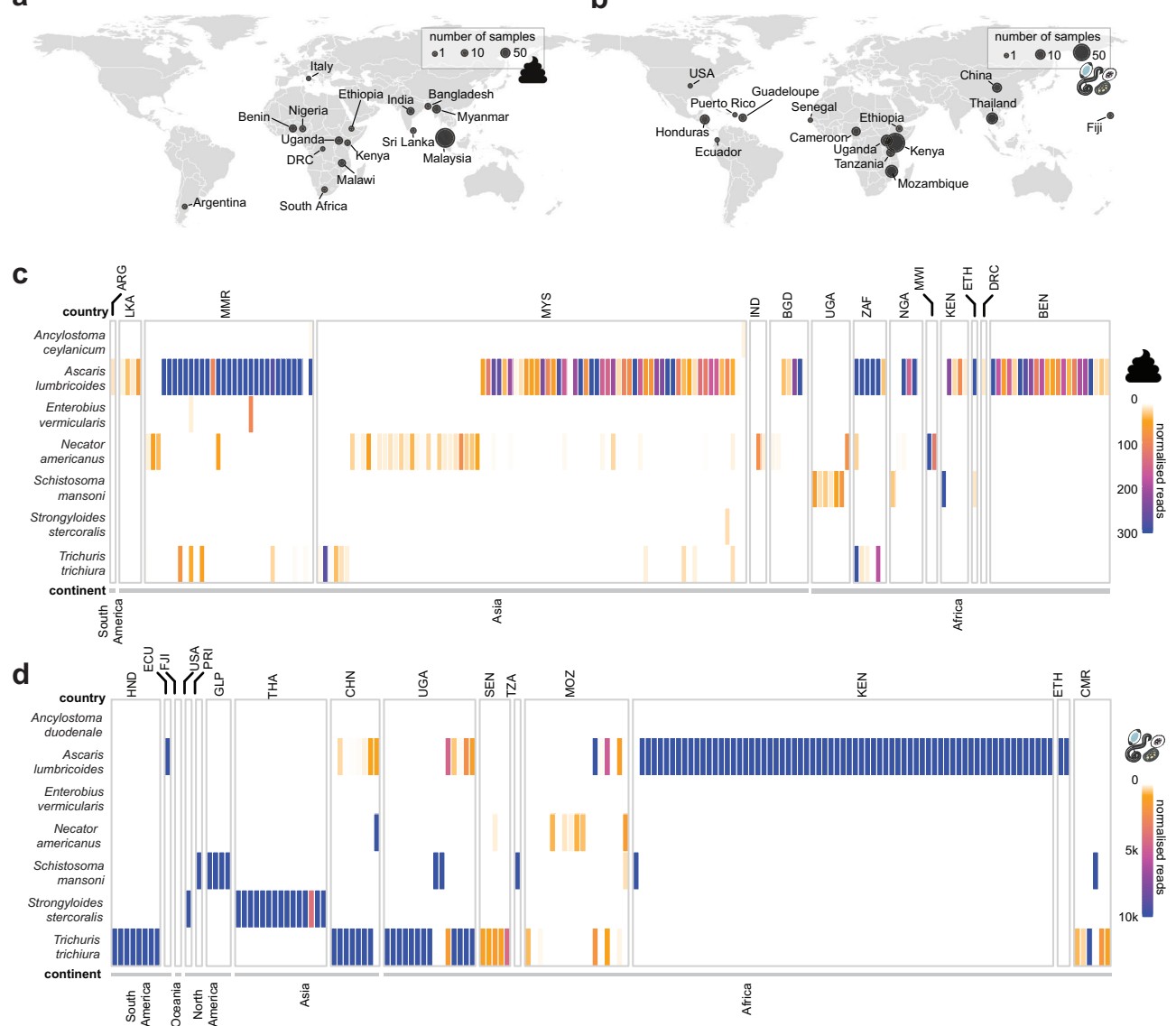

**Fig. 1 | The geographic distribution of gastrointestinal helminths detected by sequencing among worm or egg isolates and faecal samples analysed.** World maps show the countries of origin for all samples used in the study, which include (**a**) faecal samples (*n* = 842) from 15 countries and (**b**) worm and concentrated egg isolates (*n* = 158) from 15 countries. The point size on the maps indicates the number of samples from each location. Heatmaps display the relative prevalence of single and mixed-species infections of parasites detected by sequencing from (**c**) faecal samples and (**d**) worm/concentrated egg isolates, respectively. The faecal sample icon indicates faecal samples and adult worm/egg figures indicate samples from adult worms and/or concentrated worm eggs. A minimum normalised coverage threshold of 10 reads was applied, resulting in 329 samples from 27 countries reported as positive by sequencing for at least one parasite. Genomic datasets from faecal samples were plotted separately from the worm or egg datasets using a

different scale based on the obtained coverage. Colours reflect read counts, normalised by the total number of reads per sample per genome size, to achieve 'reads mapped per million reads per Mb'. Sample site abbreviations for both faecal and worm data are as follows: ARG = Argentina; BEN = Benin; BGD = Bangladesh; CHN = China; CMR = Cameroon; DRC = Democratic Republic of the Congo; ECU = Ecuador; ETH = Ethiopia; FJI = Fiji; GLP = Guadeloupe; HND = Honduras; IND = India; KEN = Kenya; LKA = Sri Lanka; MMR = Myanmar; MOZ = Mozambique; MWI = Malawi; MYS = Malaysia; NGA = Nigeria; PRI = Puerto Rico; SEN = Senegal; THA = Thailand; TZA = Tanzania; UGA = Uganda; USA = United States of America; ZAF = South Africa. Faecal, worm, and egg icons are provided by Servier Medical Art (https://smart.servier.com/), licensed under CC BY 4.0 (https://creativecommons.org/licenses/by/4.0/). Source data are provided as a Source Data file.

mapped preferentially to the *A. lumbricoides* reference from Korea were excluded from downstream population analyses. Remapping exclusively to the *A. suum* reference, followed by stringent variant calling, revealed 346 SNPs, which improved the genetic resolution of the remaining 81 samples (Supplementary Figs. 5a–c).

To measure genetic variation between and within populations, we assessed pairwise genetic differentiation ($D_{XY}$), nucleotide diversity ($\pi$), and the frequency of alleles ($freq_{allele}$) (for pooled populations; i.e., concentrated eggs or faecal samples infected with helminth eggs) or

frequency of SNPs ($freq_{SNP}$) (for individual adult worms). $D_{XY}$ showed contrasting patterns between *A. lumbricoides* and *T. trichiura* (Fig. 2a). For *T. trichiura*, we found evidence of low genetic differentiation between sub-populations of parasites within a given country and that populations from geographically close countries were more genetically similar than from more geographically distant countries (Fig. 2a, Supplementary Figs. 6a,b; mean$D_{XY}$[between] / mean$D_{XY}$[within] = 5.59x and >33x for worm and egg, respectively), with genetically distinct populations (Supplementary Figs. 6c,d) consistent with previous

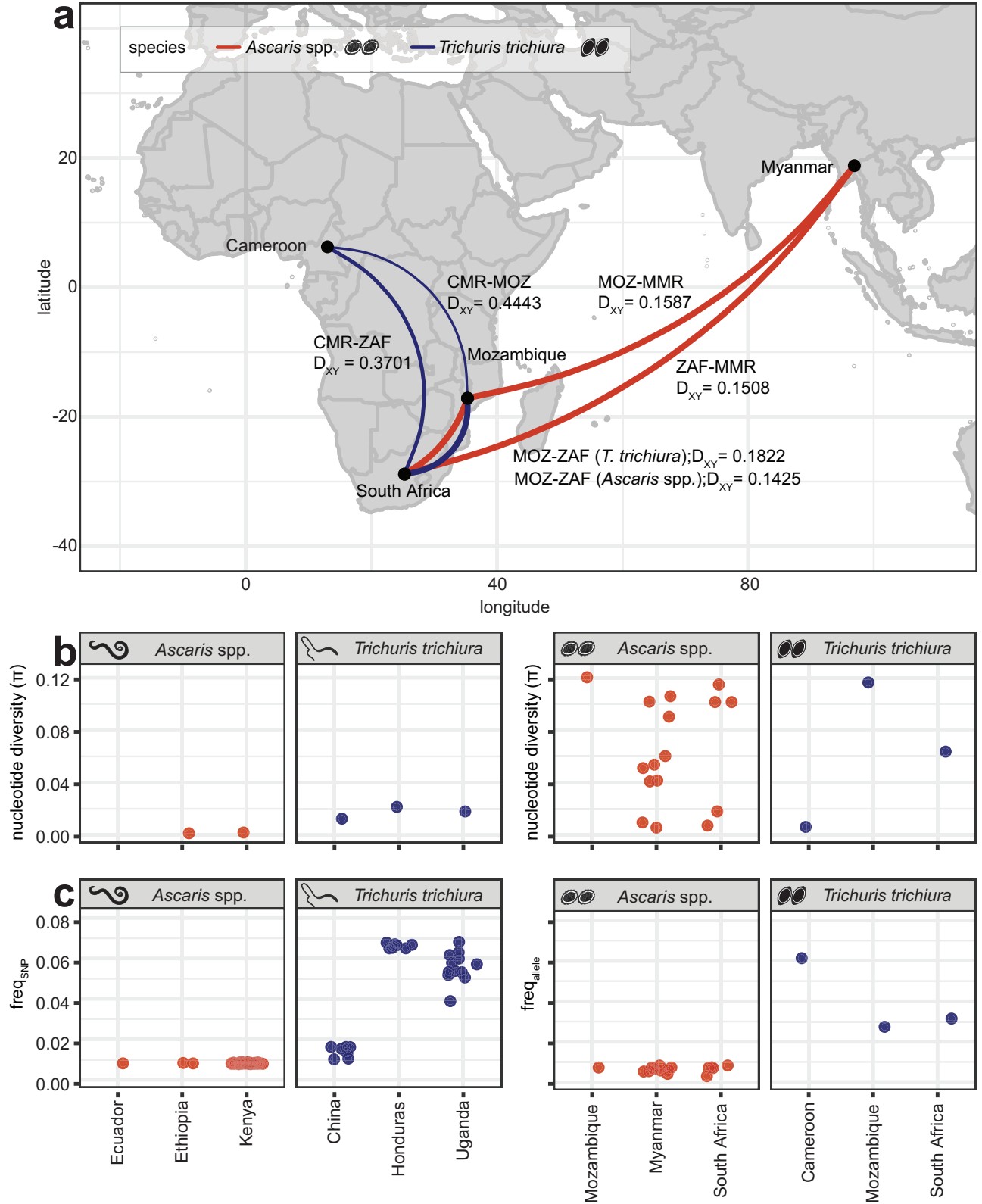

studies of *Trichuris* genetic variation[25]. By contrast, *A. lumbricoides* showed low levels of differentiation comparing within versus between country $D_{XY}$ (mean$D_{XY}$[between] / mean$D_{XY}$[within] = 2.48x and 1.47x for worm and egg, respectively) and significantly lower levels of genetic differentiation between populations despite geographic distance, even between populations from Africa and Asia (Fig. 2a, Supplementary Figs. 5a,b,c,d). The contrasting patterns between species

are likely driven by consistent differences in the patterns of genetic variation within populations as measured by nucleotide diversity (Fig. 2b) and variant frequency (Fig. 2c) in both worm and egg samples. In almost all cases, nucleotide diversity and variant frequency were higher in *T. trichiura* than in *A. lumbricoides* populations, though results for *Trichuris* were only supported by a few positive samples. This pattern of variation in *A. lumbricoides* is consistent with that of

**Fig. 2 | Comparison of genetic diversity within and between populations of *Trichuris trichiura* and *Ascaris* spp. a** Relative genetic relationships between populations for pooled samples (i.e., eggs) based on pairwise estimates of mitochondrial genome diversity (D$_{XY}$). Colour indication: blue = *Trichuris trichiura*, red = *Ascaris* spp. The thickness of the line connecting countries reflects the degree of similarity between mitochondrial genomes; thicker lines represent more substantial genetic similarity, while thinner lines represent weaker genetic similarity. The mean D$_{XY}$ per country combination was calculated, and the line size was plotted as '1- mean D$_{XY}$ per country combination'. **b** Comparison of nucleotide diversity (π) in mitochondrial genomes amongst individual worms (left) and pools of eggs from faecal samples (right) (*n* = 16 for *Ascaris* spp., *n* = 3 for *T. trichiura*) per population. **c** Comparison of the variant frequencies as SNPs in the individuals'

populations (*n* = 28 samples for *T. trichiura*, *n* = 65 for *Ascaris* spp.) and alleles in the pools (*n* = 16 for *Ascaris* spp., *n* = 3 for *T. trichiura*). The frequencies of SNPs and alleles were calculated to facilitate comparison of genetic variation among both individual worms and pools of eggs. For Ecuador, calculating nucleotide diversity was infeasible due to the availability of only a single sample. Adult worm and egg icons represent single adult worm and concentrated worm egg data, respectively. Country codes are as follows: CMR = Cameroon; MMR = Myanmar; MOZ = Mozambique; ZAF = South Africa. Faecal, worm, and egg icons are provided by Servier Medical Art (https://smart.servier.com/), licensed under CC BY 4.0 (https://creativecommons.org/licenses/by/4.0/). Source data are provided as a Source Data file.

Easton et al.[39], where very few variants were found to differentiate haplotypes from reference genomes sampled from around the world, even though the data presented here exhibit lower coverage and more sampling with higher depth would be necessary to validate such trends for both species. Collectively, these data suggest that *Ascaris* has more recently and more rapidly spread around the world, likely supported by relatively higher reproductive output and zoonotic transmission that may have been exacerbated by modern livestock transport, homogenising geographically distant populations[31]. Further targeted sampling of *Ascaris* from humans and pigs living within proximity or cohabiting in endemic regions will shed further light on zoonotic transmission and its contribution to parasite radiation.

## Nuclear diagnostic markers vary substantially in number and sequence within and between STH species

We next sought to understand the extent to which nuclear diagnostic markers, specifically those used as targets for qPCR, vary in sequence and abundance throughout the genomes of STHs. Published diagnostic repeat targets (Supplementary Data 4) were used to identify the coordinates of all similar repeat variations in each genome, retaining candidate repeats throughout the genomes with a least 90% coverage and nucleotide identity to the published target (Supplementary Data 5). Characterisation of the location of repeat arrangements in the *A. lumbricoides*, *N. americanus*, and *T. trichiura* reference genome assemblies (Supplementary Figs. 7a,b,c, respectively) revealed coordinates for (i) 125 germline tandem repeat targets (spread across 11 scaffolds) and six internal transcribed spacer (ITS) rDNA targets (across two scaffolds) for *A. lumbricoides*, (ii) 302 tandem repeat targets (across 11 scaffolds) and 15 18S rDNA targets (in one scaffold) for *T. trichiura*, and (iii) 336 tandem repeat targets (across eight scaffolds) and two ITS targets (in one scaffold) for *N. americanus* (for marker details, see Supplementary Data 4). A pairwise comparison of all repeats within each species highlighted substantial sequence diversity among repeats within each species (Fig. 3a, b, c), despite all repeats being identified using a single canonical repeat target sequence as a query. The sequence diversity among repeats suggests that not all sequences are suitable qPCR targets; only 80/125 (*A. lumbricoides*), 221/302 (*T. trichiura*), and 335/336 (*N. americanus*) sequences included the complete primer and probe binding sites necessary for amplification and detection, respectively (a list of binding sites for all oligos for all species is shown in Supplementary Data 6).

Our understanding of the repeat arrays in a genome, as observed above, is based on a single reference genome that likely contains technical variation (i.e., collapsed repeats resulting from misassembly) and biological variation (i.e., variations in repeat copy number) compared to the original isolate sequenced. To explore the latter, we estimated the coverage of each repeat per country per species, which was normalised by the coverage of exons from single-copy genes. Overall, *A. lumbricoides* (Fig. 3; n = 69/80 repeats with coverage >0) and *N. americanus* (Fig. 3; n = 49/335 with coverage >0) contained fewer repeats with more variable coverage relative to *T. trichiura* (Fig. 3f; n = 81/221 with coverage >0). We attribute this to the difference

between short-read assemblies of *A. lumbricoides* and *N. americanus* (prone to collapsed repeats) and the more contiguous long-read assembly of *T. trichiura* in which the repeat arrays are likely to be better assembled. Nonetheless, repeat coverage varied substantially between countries and repeat types, suggesting that population-specific variation in repeat copy number may play a role in the qPCR-based diagnosis of STHs.

The sensitivity and specificity of qPCR assays targeting repetitive sequences are influenced by whether the targets lie within the nuclear or mitochondrial genome. Overall, the relative copy number of nuclear repeats was higher than that of mitochondrial genomes (except for adult *A. lumbricoides*, which had a larger copy number of mitochondrial genomes than nuclear repeats, likely due to repeat loss in adults as a consequence of chromosome diminution[41]). These ratios were more consistent across species in eggs than in adults (Supplementary Figs. 8a, b). Considering that, in practice, the DNA of eggs from faeces is the primary target of the qPCR diagnostic for STHs, our data support the underlying hypothesis that nuclear repeats provide a higher degree of sensitivity (relative to mitochondrial genome targets or other single-copy genes), albeit with the caveat of greater variation of nuclear repeats within and between STH species than previously appreciated[42].

## Genetic variation within primer sites can detrimentally affect qPCR amplification of diagnostic targets

Sequence and copy number variation within and between populations prompted us to investigate whether individual genetic variants within nuclear repeats, particularly in the primer binding sites, would affect the diagnostic performance of the qPCR assays. Stringent variant filtering of nuclear-mapped data (Supplementary Data 7), keeping only variants on oligo (primer/probe) binding sites, identified 144 SNPs (n = 68 samples) for *A. lumbricoides*, 508 SNPs (n = 29 samples) for *T. trichiura*, and 123 SNPs (n = 4 samples) for *N. americanus*. To test the effect of these SNPs on qPCR assays, we identified repetitive sequences for each species that (i) contained SNPs that varied substantially in frequency between populations and (ii) contained SNPs within five bp from the 3' end of either of the primers that would result in a priming-mismatch. From these variants, we selected three tandem repeats from each of *A. lumbricoides* (Fig. 4a) and *T. trichiura* (Supplementary Fig. 9a) for validation. No variants meeting those criteria were found for *N. americanus*.

To assess whether these variants in the binding sites might impact primer annealing and amplification, we evaluated the impact of these variants in established qPCR assays[20,21]. This was achieved using a serial dilution series of cloned target sequences, each containing either the wild-type or mutant base (Supplementary Data 8). For *A. lumbricoides*, the presence of the variant in both repeats 1 and 2 (Al_SNP_1 and Al_SNP_2; both A-to-G variants, both in scaffold_3) did not reduce the efficiency of the qPCR assay, whereas, in repeat 3 (Al_SNP_3; T-to-G variant), the efficiency decreased to 83.9% (Fig. 4b). However, the presence of variants in the primer binding sites resulted in >1000-fold loss of amplified product in repeat 2 (Al_SNP_2), followed by >500-fold

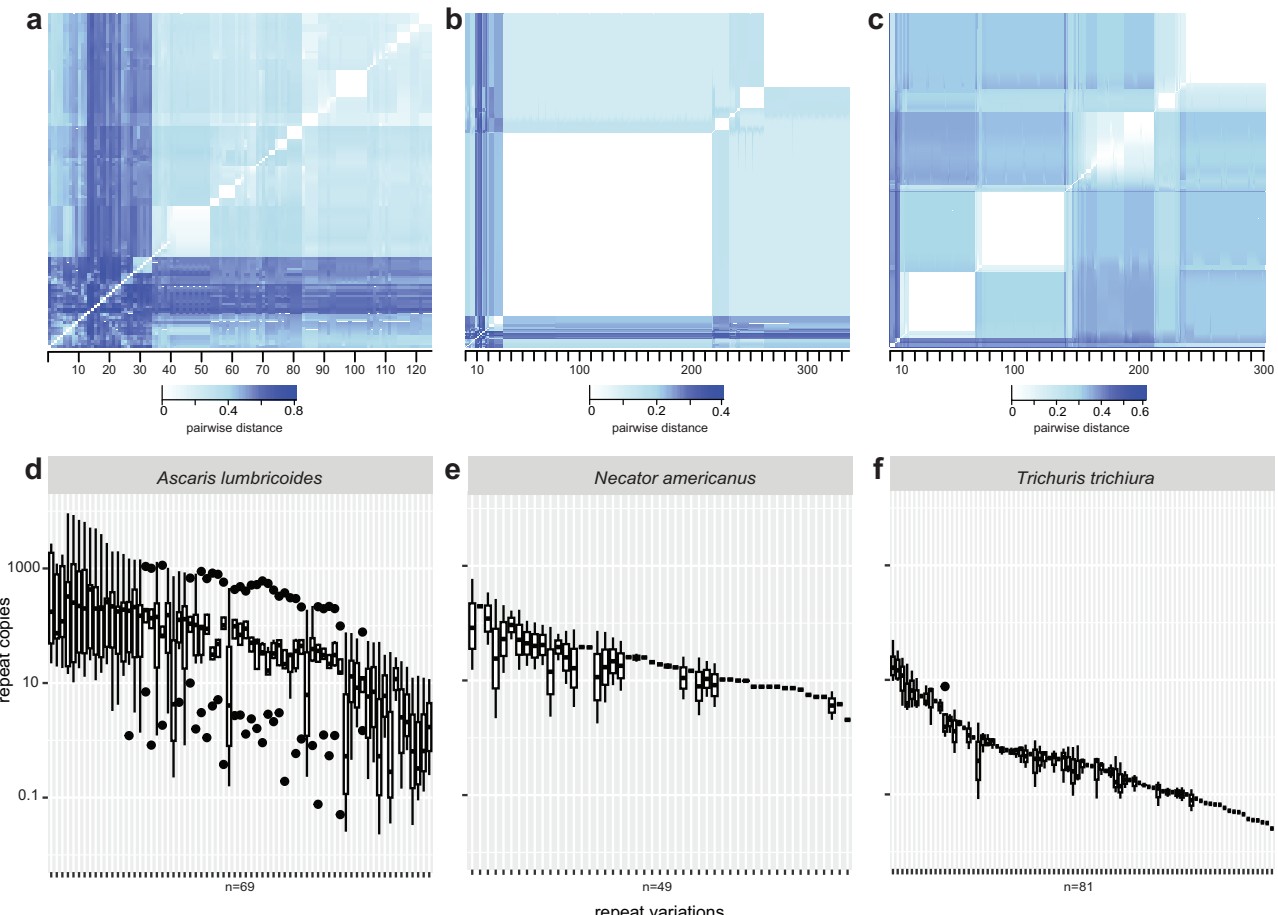

**Fig. 3 | Genomic diversity and differential coverage of repeat classes used as diagnostic targets. a, b, c,** Repeat diversity was determined by searching the canonical repeat units using *nucmer* with 90% nucleotide similarity and 90% sequence coverage against the genome assemblies of (**a**) *Ascaris lumbricoides*, (**b**) *Necator americanus* and (**c**) *Trichuris trichiura* as a reference. In **a**, **b**, and **c**, the heatmaps show the pairwise distance calculated as the sum of squares of a nucleotide similarity matrix derived from *ClustalOmega*-aligned repeat sequences for each species, where lighter colour (white) on the colour scale reflect a stronger degree of similarity between two sequences. In **d**, **e**, and **f**, Genome coverage per repeat per country was determined by *bedtools multicov* (with minimum overlap 0.51) in merged-by-country BAM files (filtered raw reads > ten reads) against the genome assemblies of (**d**) *A. lumbricoides*, (**e**) *N. americanus*, and (**f**) *T. trichiura*. Coverage is expressed as 'repeat copies,' calculated by dividing the original repeat coverage by the mean per-country single copy exon coverage. The central box represents the interquartile range, and the whiskers represent the data's first and third quartiles. The median is shown as a line through the centre of the box. The whiskers extend from the edges of the box to the smallest and largest values within 1.5 times the interquartile range (IQR) from Q1 and Q3, respectively. Only repeats containing both forward and reverse primers and probe binding sites were included. Source data are provided as a Source Data file.

loss in repeat 1 (Al_SNP_1), and >50-fold loss in repeat 3 (Al_SNP_3) (Fig. 4c). We observed similar patterns for *T. trichiura*. The presence of the SNP in repeat 1 (Tt_SNP_1; T-to-G variant), repeat 2 (Tt_SNP_2; G-to-C variant) and repeat 3 (Tt_SNP_3; T-to-A variant) prevented any amplification of the target (Supplementary Fig. 9b). However, the variants occurred in regions where the primers and probe had several mismatches with both the wildtype and mutant templates (five-to-six on average) and, therefore, we were unable to test the effect of the mutation alone without accounting for the mismatches. The impact of the mismatches alone revealed >2.1e7-fold loss in product amplification in repeat 1, compounded by an additional 2000-fold loss due to the mutation (SNP 52; Supplementary Fig. 9c). In repeats 2 and 3, the mismatches resulted in a > 4.5e4- and 1.6e4-fold loss, respectively, and the presence of SNP caused an additional 4.6e6 (Tt_SNP_2) and 1.3e6 (Tt_SNP_3) fold-loss, respectively.

Considering the high prevalence of mismatches between the published primers and the genome assemblies, we sought to determine whether this discrepancy was specific to the genomes used or if the primers and the repeats from which they were designed were broadly different. Detection of the canonical repeats (without any primer mismatches or SNPs) in the other two available genomes for *A.*

*lumbricoides* (A_lumbricoides_Ecuador_v1_5_4; WormBase ParaSite [WBPS]) and *T. trichiura* (TTRE3.0; WBPS) revealed no *Ascaris* repeats with 100% nucleotide identity (of 26 repeats with 90-100% nucleotide identity) and in *T. trichiura*, only three matches with 100% nucleotide identity were detected (of 302 repeats with 90–100% nucleotide identity). Considering that these assays were initially designed based on a consensus sequence[20,21,43], it is likely that these mismatches reflect false variants. Given that these false variants are found in the primers themselves, these assays likely perform suboptimally due to the priming of mismatched targets.

These results demonstrate that the occurrence of variants, which we have identified as variable in frequency among globally distributed STH populations, in qPCR targets used as diagnostics can significantly impact the detection of STHs. In practical terms, these variants caused a 3- to 10-cycle change in qPCR, which would lead many samples, especially those with low-intensity infections, to yield false-negative results for STHs.

## Discussion

Soil-transmitted helminths impact a large portion of the global population, perpetuating poverty and disease in some of the most

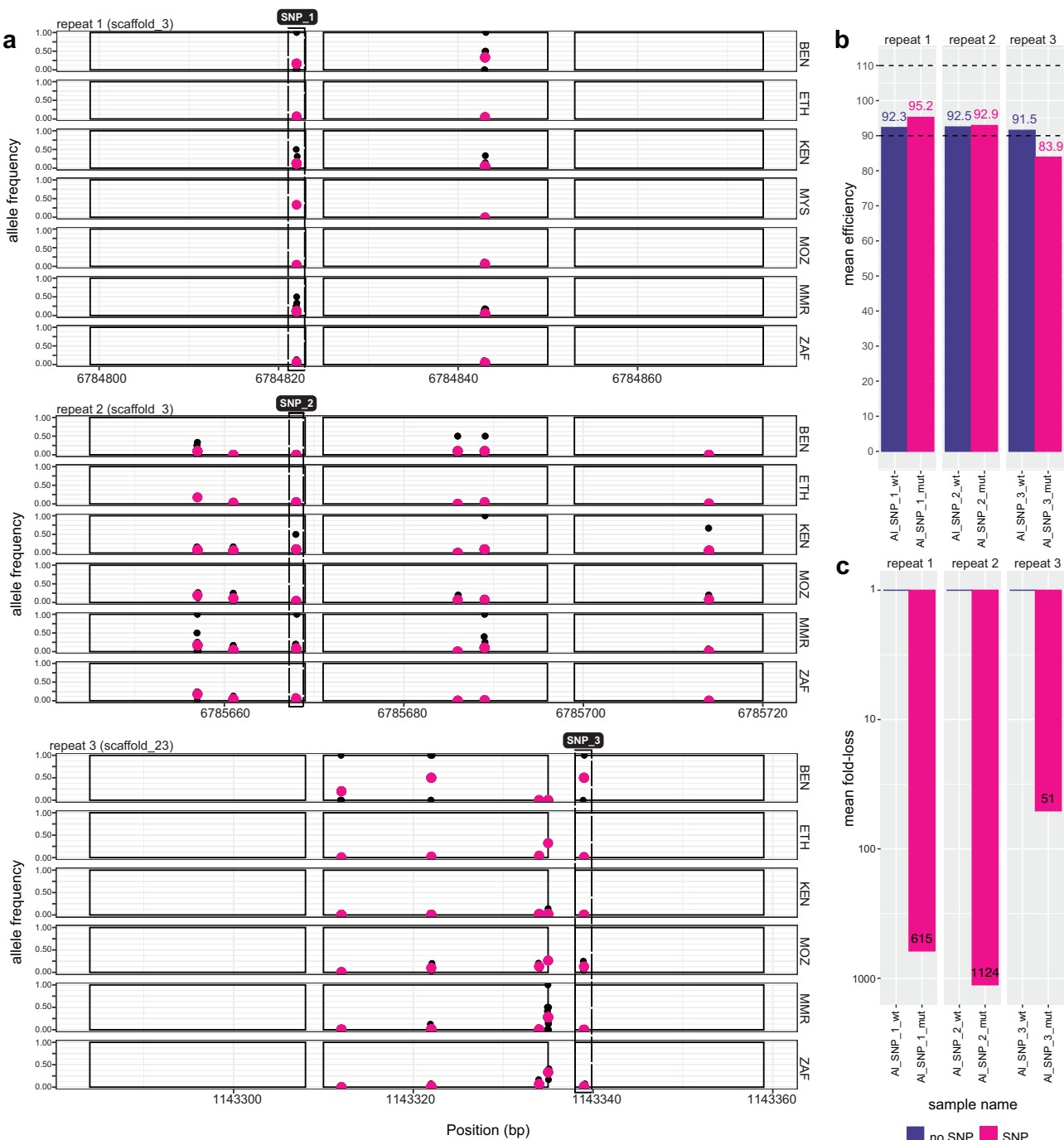

**Fig. 4 | Presence, distribution, and impact of genetic variation within diagnostic qPCR targets of *Ascaris lumbricoides*. a** Shown are the genomic coordinates of three repeats - repeat 1, 2, and 3 - highlighting primer and probe binding sites (solid rectangles) used in the qPCR diagnostic test, the position of genetic variants (x-axis) - either individual samples (black points) or mean across samples (pink points) - and their frequency within each country (y-axis). Putative qPCR-disruptive variants found at the 3' end of the primer binding sites are depicted by dashed rectangles. **b** qPCR efficiencies were determined by generating standard curves of five serial dilutions (100 pg/µl to 10 fg/µl) on each of the repeats in the absence (wildtype; wt) or presence of the SNP (mutant; mut). The mean value of three replicates per concentration is shown. The 90-110% dashed lines show acceptable qPCR efficiency cutoffs. **c** The mean fold-loss was calculated to assess the effect of the SNP in qPCR quantitation and product loss due to late amplification. The mean fold loss of the mutant is relative to the wildtype repeat, within each assay. The mean normalised $C_q$ difference was estimated from all serial dilutions. Country codes are as follows: BEN = Benin; ETH = Ethiopia; KEN = Kenya; MMR = Myanmar; MOZ = Mozambique; MYS = Malaysia; ZAF = South Africa. Source data are provided as a Source Data file.

neglected communities worldwide. Effective and sustainable control of STHs requires a robust understanding of how parasite populations change over time. Here, we have addressed three key aspects of genomic epidemiology relevant to STH control, characterising (i) complex samples containing one or more helminth species, (ii) genetic

diversity and connectivity of parasite populations, and (iii) the impact of genetic variation between STH populations on the performance and efficiency of conventional molecular diagnostics such as qPCR.

Measures of genetic diversity from parasite populations have the potential to provide vital information for STH epidemiology, from

their natural transmission patterns within and between populations, through population decline in response to drug treatment (including drug resistance) to recrudescence after treatment termination. Although an increasing number of studies have focused on helminth genetic diversity[25,32,38,39,44], very little is known about the genetic diversity of STHs throughout their geographical range. We have identified contrasting patterns of genetic variation and connectivity between globally distributed STH species. We note that due to the mixed nature of the sample cohort, it was difficult to directly compare the pooled egg or faecal samples with the individual worms, which limited our ability to assess population dynamics at a higher granularity. We also acknowledge that low-coverage sequencing poses some limitations. While the approach was specific, its sensitivity was lower than that of a targeted approach[45], which likely explains the higher frequency of *Ascaris* infections due to the greater biomass of eggs and the lack of *N. americanus* data, despite its high prevalence. Moreover, DNA sequencing from faecal samples meant that most of the reads were not helminth-derived and, therefore, discarded. Nonetheless, we were able to detect population genetic signatures of *T. trichiura* that are consistent with increasing genetic differentiation as a function of increasing geographic distance, likely due to limited dispersal. We also gained insights into likely zoonotic infections between pig-infective *A. suum* and human-infective *A. lumbricoides*. Although all samples analysed here were obtained from humans, there was an overwhelming bias towards *A. suum*-derived genetic diversity, leading us to question the evolutionary background of the original reference sample. Considering that *N. americanus* and *T. trichiura* are also zoonotic[25,44,46–48], genetic analysis of both human and animal reservoirs (monkeys, baboons, macaques for *Trichuris* and chimpanzees, gorillas for *Necator*) will become increasingly important for precisely understanding the populations targeted for control. These analyses will benefit from new high-throughput molecular tools to characterise parasite diversity, including more sensitive sampling of nuclear rather than mitochondrial genomes, and robust, consistent sampling approaches to enhance the representativeness and, consequently, the genetic and geographic resolution of STH population genomics.

Understanding genetic variation, ranging from single nucleotide polymorphisms to copy number variation of repetitive sequences, relies on reference genome resources that differ in contiguity and completeness for STH species[49]. Our nuclear genome analyses revealed the position, number, and nucleotide identity of these repeats within the genomes of *A. lumbricoides*, *N. americanus*, and *T. trichiura*. However, our interpretation of these regions will change as genome assemblies improve, as evidenced by the lower variation in copy number and higher representation of repeats in the newer, PacBio-derived *T. trichiura* genome. The availability of new genomes based on long-read sequencing will help resolve the complexity of difficult-to-assemble repetitive regions of the genome (e.g.[50]) and, therefore, be more robust in identifying genetic variants in diverse helminth populations. New long-read assemblies should also include diverse isolates to capture structural and geographically specific diversity that may be missed by short-read sequencing alone.

The development of molecular approaches to diagnose and monitor STH infections offers significant potential for increased sensitivity, specificity, and throughput compared to conventional microscopy approaches. As such, qPCR-based diagnostics targeting repetitive sequences have been used in large-scale epidemiological and clinical trials to confirm elimination or monitor programmatic progress[15–17,51]. A list of qPCR assays targeting genes or non-coding repeats for the diagnosis of STHs is provided in Supplementary Data 4. Genetic variation in diagnostic targets may impact the monitoring of STHs by control programmes. If not accounted for, there is a risk of undiagnosed false-negative infections if qPCR is applied in programme settings. We demonstrate that, under in vitro controlled conditions, single nucleotide genetic variants can significantly influence the ability

of qPCR to detect these diagnostic repeats. We acknowledge that our in vitro assays are extreme scenarios and assume that all targets in a genome would be identical and perform equally poorly in the presence of the variant. Our analyses of repeat diversity show this to be false in reality. However, detecting the canonical repeat sequences within the reference genomes was challenging, suggesting that these established assays may perform sub-optimally. Redesigning diagnostic targets while accounting for genome-wide population genomic data will improve the assay by avoiding known variants and may enable the selection of more conserved targets that are less sensitive to disruptive genetic variants. Moreover, validation of new assays using a cohort of geographically diverse samples that are representative of the continent, region, and/or country would provide a greater understanding of the assay's strengths and weaknesses and, potentially, greater confidence in its application in new populations.

Our study emphasises the importance of sampling a wider range of human and nonhuman hosts globally to improve our understanding of the genetic makeup and evolution of STHs over time and space. Further work to define the genetic connectivity of STH populations may enable the definition of transmission zones and networks, providing a more precise means to prioritise control efforts. Our analyses provide a clear rationale for a more comprehensive evaluation and validation of repetitive sequences currently used as targets in molecular diagnostics. Collectively, these data and subsequent studies will form the foundations for the genomic epidemiology of all STHs and their sustainable control as a public health problem.

## Methods
### Origin of samples and datasets
The complete list of samples, country of origin, and accession numbers is provided in Supplementary Data 1. No sample size calculations were used in the study design. Samples were chosen to maximise the number of populations in geographically distinct regions of the world. A total of 1000 samples were used in the study; 850 (from 19 countries) were publicly available short-read shotgun metagenomics datasets obtained from the European Nucleotide Archive (ENA) from worm and faecal/egg samples known to have STH infections confirmed by either microscopy or qPCR, and 150 (13 countries) were DNA samples (from worm and faecal material) that underwent low-coverage genome sequencing, which were selected from a larger cohort of samples based on positive infections from microscopy data.

The ethical approvals for the collection of whole genome sequencing data from 150 samples across various countries followed rigorous protocols, as follows: Argentina ($n = 1$), approved by the Colegio de Médicos de la Provincia de Salta and BCM (protocol number H-34926); Bangladesh ($n = 10$), icddr,b (Ref: PR-14105), University of California Berkeley (Ref: 2014-08-6658), and Stanford University (Ref: 27864); Benin ($n = 25$), approval came from the IRCB (Ref: 002-2017/CNERS-MS) and the University of Washington (Ref: STUDY00000180) and the Data Safety and Monitoring Committee (DSMC); Democratic Republic of the Congo ($n = 1$) approved by Democratic Republic of the Congo/ Ghent University (M104; Catholic University of Bukavu, Ref: UCB/CIE/NC/016/2016) and the Ministry of Public Health ([Ref: 062/CD/DPS/SK/2017]); Ethiopia ($n = 3$) approved by Ghent University (Ref: B670201627755 and PA2014/003) and Jimma University (Ref: RPGC/547/2016); Fiji ($n = 1$) from the Natural History Museum, London (Ref: 2012.11.19.1); India ($n = 25$) approved by the Christian Medical College (Ref: 10392) and the University of Washington (Ref: STUDY00000180) and by the DSMC; Kenya ($n = 7$) approved by KEMRI (Ref: SSC #1820); Malawi ($n = 25$) approved by London School of Hygiene and Tropical Medicine (Ref: 12013), the College of Medicine (Ref: P.04/17/2161), and the University of Washington (Ref: STUDY00000180) and the DSMC; Myanmar ($n = 32$) approved by Imperial College London (Ref: 17IC4249 and 17IC4249 NoA1); Nigeria ($n = 11$) approved by the Kebbi State Ministry of Health

provided ethical approval (Ref: 105:23/2021); South Africa ($n = 7$) approved by the University of KwaZulu-Natal (Ref: BF029/07); Sri-Lanka ($n = 1$) approved by the Faculty of Medicine, University of Peradeniya (Ref: 2015/EC/58). More specific details regarding the samples from each country are available in the Supplementary Information.

## DNA extraction and high-throughput sequencing

The details below include DNA extraction and library preparation for the 150 samples that were de novo sequenced in this study. Details on the remaining 850 genomic datasets (per country, including relevant ethical approvals and accession numbers) are provided in the Supplementary Information.

As a consequence of collating a selection of diverse samples from multiple countries, multiple different extraction kits were inevitably used to isolate DNA from the 150 faecal or worm samples (Supplementary Data 9) that were sequenced as part of this study. The faecal samples and/or concentrated eggs from Argentina ($n = 1$), Bangladesh ($n = 10$), Myanmar ($n = 32$), Kenya ($n = 7$), and South Africa ($n = 7$) were processed using the FastDNA Spin Kit for Soil (MP Biomedicals) and a high-speed homogeniser with modifications, including an extraction control[51]. The sample cohorts from Benin ($n = 25$), Malawi ($n = 25$), and India ($n = 25$) were extracted as described elsewhere using a 96-well plate shaking system and a KingFisher Flex purification system with the MagMax Microbiome Ultra Nucleic Acid Isolation Kit[19]. The faecal samples from Ethiopia ($n = 1$) and the samples from Democratic Republic of the Congo ($n = 1$) were processed using the QIAsymphony DSP Virus/Pathogen Kit (QIAGEN) with modifications, including the use of a high-speed homogeniser (Vlaminck et al.,). The samples from Sri Lanka ($n = 1$) were processed using the PowerSoil DNA Isolation kit (Mobio, QIAGEN), including a negative/blank extraction control. The samples from Nigeria ($n = 11$) were extracted with the ZymoBIOMICS kit for DNA extraction (Zymo Research). The worm samples from Fiji ($n = 2$) and Ethiopia ($n = 2$) were extracted using the Isolate II Genomic DNA extraction kit (Bioline, Meridian Bioscience).

Sequencing libraries were prepared from the DNA extractions using the Plant and Animal Whole Genome Sequencing pipeline (Novogene, Hong Kong). One hundred fifty samples were sequenced on a single lane of an Illumina NovaSeq 6000 (Novogene, Hong Kong) using $2 \times 150$ bp paired-end chemistry, generating approximately ten gigabases (Gb) of sequencing data per sample. Due to the initial low data yield of 10 samples, the libraries were sequenced further on two additional sequencing lanes. In total, 149 of 150 samples generated sequencing data for further analysis. Individual sample accessions are described in Supplementary Data 1.

## Raw read processing and mapping to reference genomes

Reference FASTA files representing the mitochondrial genomes and whole-genome assemblies of helminths previously reported to be found in the faeces of human hosts were collated from WBPS (release WBSP17[52]) or NCBI (Supplementary Data 2). An improved, recently published genome assembly of *T. trichiura*[25] and the most recent genome of *A. lumbricoides*[39] were used instead of the assembly versions available from WBPS. The reference sequences were combined to generate a metagenomic reference set for each of the mitochondrial or whole genome sequence sets. By doing so, metagenomic sequencing reads composed of one or multiple species could map competitively and specifically to the reference genome of the species from which they were derived, enabling the generic classification of any sample.

Reference FASTA files were indexed using the Burrow-Wheeler Aligner (BWA) before trimmed sequencing reads (using *Trimmomatic*, v.0.39-2) were mapped using BWA-MEM (v.0.7.17-r1188). Unmapped reads were filtered using *samtools* (v.1.6) *view*, retaining reads >80 bp in length. Mapped reads were further filtered to remove hard clipped (based on CIGAR pattern) reads using *samclip* (v.0.4.0), after which unique alignments were retained from the sorted BAM files using

*sambamba* (v.0.8.2[53]). The remaining soft-clipped alignments were filtered further by removing any reads with 'S' (indicating a soft-clipped alignment) on the CIGAR column of the BAM files. Lastly, duplicated reads were marked and removed from the BAM files using *Picard MarkDuplicates* (v.picard-2.18.29-0).

The number of mapped reads per sample per species was analysed using *bedtools multicov* (v.2.30.0) on filtered BAM files. To compare diverse samples, read count normalisation was performed by dividing the number of reads mapped by the total number of reads per sample and then dividing by the mitochondrial genome size in Mb. A minimum mean coverage of 5x was selected for downstream analysis. Co-infections across all species and samples were visualised with the *UpSetR* package (v.1.4.0) in R[54].

## Mitochondrial genome variant calling

Variant calling was performed using *bcftools mpileup* (v.1.16). Two VCF files were generated, one containing only variant sites (used for pooled data, downstream, filtered for both min and max number of alleles = 2) and a second VCF that uses both variant and invariant sites (used for genotype-based population analysis in the individual worm data, filtered only for max alleles = 2). The only-variants-VCF file was further filtered to include only SNPs within coding genes (*vcftools --vcf* VCF *--bed* GENE.bed). A data-driven approach was employed to filter variants by removing the tails (lower 10%) of variant distributions based on QUAL metrics. Variants were further filtered to ensure (i) a maximum of two alleles, (ii) unique variants, (iii) only SNPs to be retained (any insertions/deletions were removed), and (iv) a per sample missingness >0.7. The number of SNPs surviving each filter (and the number of individuals) is shown in Supplementary Data 3.

## Population structure analysis

Broad-scale genetic relatedness between samples and populations was explored by analysing mitochondrial genome allele frequencies using principal component analysis from the R package *pcaMethods* (v.pcaMethods_1.90.0[55]). Specifically, Bayesian principal component analysis (BPCA) was used to account for missing values in the dataset. Due to the inclusion of both pooled and individual samples in the analysis, allele frequencies were used rather than genotypes. To ensure that any existing population clustering was not attributed to differences in coverage or missingness in the data, each BPCA plot was coloured by normalised coverage and shaped by missingness in the data (total missingness present < 0.3). Genetic differentiation between samples was performed using *Grenedalf* (v.0.3.0[56]) for the pooled (faecal or concentrated eggs) samples and *pixy* (v.1.2.7.beta1[57]) for the samples originating from individual worms.

## Mitochondrial genome-wide genetic diversity analyses

Samples were split and analysed in two groups, either (i) pools or (ii) individuals, depending on whether they originated from faecal samples (including concentrated eggs) or individual worms, from which nucleotide diversity ($\pi$) per population and pairwise genetic divergence ($D_{XY}$) were determined.

In a pooled sequencing context, $\pi$ is the average pairwise difference between all reads in a sample; higher $\pi$ values indicate a rich pool of genetic variation within a population, whilst lower $\pi$ values indicate lower genetic diversity, possibly due to population bottlenecks, founder effects, and strong selection pressures. $D_{XY}$ reflects the absolute divergence in $\pi$ between two populations; higher $D_{XY}$ values indicate greater genetic divergence, reflecting potential isolation or selection pressures that have shaped the populations, whereas lower $D_{XY}$ values suggest more genetic similarity, possibly due to recent shared ancestry or ongoing gene flow.

For the pools, $\pi$ per population and pairwise $D_{XY}$ were determined using *Grenedalf* (*grenedalf diversity --filter-sample-min-count 2 --filter-sample-min-coverage 1 --window-type chromosomes --pool-sizes 1000*

--vcf-path; grenedalf fst --write-pi-tables --filter-sample-min-count 2 --filter-sample-min-coverage 1 --filter-region-list --window-type chromosomes --pool-sizes 1000 --method unbiased-hudson). For the individuals, *pixy* was used (*pixy --stats pi dxy --vcf --chromosomes --populations --window_size 20000 --bypass_invariant_check 'yes'*). For the calculation of π and $D_{XY}$ for the individuals, an all-sites VCF containing both variant and non-variant sites was used, whereas, for the pooled data, the original BAM files were used to avoid intrinsic biases in the VCF file from the variant caller (originally designed to be used with diploid data).

For this study, a mean $D_{XY}$ value per country comparison was calculated. Variant frequency was also calculated for both pooled ($freq_{allele}$) and individual ($freq_{SNP}$) samples, as the number of variant positions (AC = 1) divided by the total number of positions as a metric to compare individuals to pools.

## Characterisation of the location, structure, and nucleotide identity of diagnostic repeats

A list of all the PCR-based diagnostic targets assessed in this study is provided in Supplementary Data 4. The different variations of each target in each genome assembly were assessed with *nucmer* (*mummer* package, v.3.23[58]), allowing for a minimum nucleotide identity of 80% and cluster size of 15. The results were filtered further for 90% coverage and 90% nucleotide identity, and the FASTA sequences from all similar targets were obtained using *bedtools getfasta* (v.2.30.0), aligned using *ClustalOmega* (*msa* package, v.1.30.1, R) and were visualised using *Heatmap.2* (package *ComplexHeatmap*, v.2.15.4) in R. A list of all the similar targets and their respective genome coordinates can be found in Supplementary Data 5. To visualise the location and coverage of each target, each genome was split into 10 kb windows (with *bedtools makewindows* (v.2.30.0)), and the coverage was calculated using *bedtools coverage* (v.2.30.0), the results of which were plotted in R.

The number of primer and probe binding sites in each genome was assessed using *seqkit locate* (v.2.7.0), allowing up to five mismatches between the oligo and target sequences. Supplementary Data 6 lists the coordinates of the primer and probe binding sites across all genomes. The intersection between repeats and oligo binding sites was assessed with *bedtools intersect* (v.2.30.0) and the resulting bed file was used to filter the downstream variants.

## Repeat coverage per country

Repeat coverage was determined using *bedtools multicov* (v.2.30.0) (with a minimum overlap of 0.51) in merged-by-country BAM files. Values were normalised by the length of each repeat and the total number of reads. *BUSCO* (v.5.6.1[59]) was used to obtain the coordinates of single-copy exons and assess coverage in the country-merged BAM files, and a mean value per country was calculated. To obtain repeat copies per country, the original repeat coverage was divided by the mean per-country single copy exon coverage and was visualised in R (v.4.2.2).

A similar approach was followed for the mitochondrial genomes. Raw read counts (per country) were divided by the length of the mitochondrial genome, after which they were further normalised with single-copy exon coverage to obtain mitochondrial genome copies per country. Mitochondrial genome copies and nuclear repeat copies were directly compared at the country and life-stage levels (worms vs. eggs). Statistical analysis was performed in R (v.4.2.2), using the Kruskal-Wallis test to assess the difference between the nuclear and mitochondrial copies, with a p-value of less than 0.05 considered statistically significant.

## Variant calling in repetitive sequences

Variant calling was performed using *bcftools mpileup* (v.1.16), generating a single VCF file per species for all targets tested. Each VCF file was filtered for the minimum and maximum number of alleles and unique SNPs (indels were excluded). Only positions within binding sites of the primer and probe oligos were retained. SNPs were filtered for quality, summed depth for alternate and reference alleles, and sample missingness (SNPs with values below the 10th quantile were removed for all the above criteria). Different degrees of per-site-missingness were also explored, and a threshold of 0.7 was selected. The number of SNPs surviving each filter (and the number of repeats) is shown in Supplementary Data 7.

## SNP selection for validation

For SNP validation, we focused on variants in the 3' end (up to 5 bp) of the qPCR oligos with variable frequencies (and higher allele depth) in most countries. Three SNPs per species were selected for testing. To accurately assess the effect of the SNP on qPCR amplification of the repeat, we tested the repeats containing the identified SNPs along with the wild-type version of the repeat (without the SNP). To do so, we synthesised (i) the wild type (wt) repeat target (no SNP present) (ii) and the mutated (mut) repeat targets (with the SNP) in pIDTSMART-AMP plasmids (Integrated DNA Technologies, UK). The complete sequences tested are listed in Supplementary Data 8. We also tested the published canonical repeats from which the original assays were designed for comparison.

## Assessing the impact of variants using qPCR standard curves

To determine the effect of variants on the qPCR amplification efficiency, all targets were assessed by generating a 5-fold serial dilution standard curve, in triplicate, from 100 pg/μl to 10 fg/μl. qPCR assays were set up for all targets using TaqPath ProAmp Master Mix (Applied Biosystems) in a final volume of 7 μl, containing 5 μl of master mix (reagent mix, forward and reverse primers, probe and nuclease-free water) and 2 μl of template DNA. To amplify the *A. lumbricoides* targets, the forward primer (CTTGTACCACGATAAAGGGCAT; 125 nM final concentration), reverse primer (TCCCTTCCAATTGATCATCGAATAA; 500 nM), and probe (TCTGTGCATTATTGCTGCAATTGGGA; 125 nM) were used. To amplify the *T. trichiura* targets, the forward primer (GGCGTAGAGGAGCGATTT; 62.5 nM), reverse primer (TACTACCCATCACACATTAGCC; 250 nM), and probe (TTTGCGGGCGAGAACGGAAATATT; 125 nM) were used. The probes contained a 5'-FAM label (double-quenched ZEN/3IABkFQ chemistry; IDT Technologies). Cycling was performed on a ThermoFisher StepOnePlus instrument, using an initial 95 °C for 10 min followed by 40 cycles of 95 °C for 15 sec for denaturation and 59 °C for 1 min for annealing and extension.

From the standard curve data, amplification efficiency values between 90% and 110% were considered to be within an acceptable range, and outlier values of higher or lower efficiency were identified. Mean fold-change was measured as $2^{abs((wild\_type\_Cq * wild\_type\_efficiency) - (mutated\_Cq * mutated\_efficiency))}$ to assess the effect of the SNP in qPCR quantitation and product loss due to late amplification.

## Reporting summary

Further information on research design is available in the Nature Portfolio Reporting Summary linked to this article.

# Data availability

The sequencing data generated in this study have been deposited in the European Nucleotide Archive (ENA) under project accession code PRJEB90452. Publicly available sequencing data was downloaded from ENA from the following studies: PRJEB3054, PRJEB2679: [https://www.ebi.ac.uk/ena/browser/view/PRJEB2679], PRJEB44010, PRJNA304165, PRJNA847183, PRJNA511012, PRJNA797994, PRJEB53235, PRJEB31375, PRJNA602131, and PRJEB2679 [https://www.ebi.ac.uk/ena/browser/view/ERR066168]. Individual sample accessions and their associated study information for all data generated and data downloaded from public archives are described in Supplementary Data 1 and

Supplementary Information. Reference genomes used for read mapping and comparative genomics are described in Supplementary Data 2. Source data are provided with this paper.

## Code availability

The code used in the study is available via GitHub at https://github.com/MarinaSci/Global_skim_analysis and is archived under a stable (https://doi.org/10.5281/zenodo.15639714) at Zenodo[60].

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

## Acknowledgements

The authors would like to thank John M Colford Jr for the support providing the samples from Bangladesh, Marisa Juarez and Paola A. Vargas for their help with egg collection and logistical support during the field sample processing in Orán, Argentina, the NHM Molecular Labs, especially Ranbir Bailey, Elena Lugli and Claire Griffin, for access and logistical support, Stefano Colombo for help with DNA extractions on the samples from Uganda, and the Helminth Genomics group at the Wellcome Sanger Institute for insightful discussions throughout this project. Sequencing of samples was supported by the Bill & Melinda Gates Foundation (USA; OPP1129535) and the Wellcome Trust (UK; 206194). MP is supported by a Harding Distinguished Postgraduate Scholarship held at the University of Cambridge (UK). JBC is a Chan Zuckerberg Biohub Investigator. MM was supported, under the guidance of SRA, by a PhD studentship at The Tamil Nadu Dr. M.G.R. Medical University, Chennai. SRD is supported by a UKRI Future Leaders Fellowship (UK; MR/T020733/1) and the Wellcome Trust (UK; 206194). Samples provided by Ghent University (Belgium) were part of field trials supported by the Bill & Melinda Gates Foundation (USA; OPP1120972). Samples from Kenya were collected with financial support from the University of Georgia Research Foundation, Inc., which was funded by the Bill & Melinda Gates Foundation (USA; INV-010428) for the SCORE Project. Samples from Mozambique were collected by the Centro de Investigação em Saúde de Manhiça (CISM), which is supported by the Government of Mozambique and the Spanish Agency for International Development Cooperation (AECID) and Institut de Salut Global de Barcelona (ISGlobal), which receives support from the grant CEX2018-000806-S funded by MCIN/AEI/10.13039/501100011033, and support from the Generalitat de Catalunya through the CERCA Program. For the purpose of Open Access, the authors have applied a CC BY public copyright Licence to any Author Accepted Manuscript version arising from this submission.

## Author contributions

M.P., D.T.J.L. and S.R.D. conceived and designed the study; S.R.D., D.T.J.L., A.W., and C.C. co-supervised the study; S.R.A., O.A., R.M.A., R.B., J.B.C., M.C.P., N.R.C., D.C., R.O.C., P.C., A.C., J.D., S.G., J.G., B.G.P., E.H., M.I., T.P.J., K.K., E.F.K., A.J.K., B.L., A.J.L., A.M., I.M., M.M., M.M.V., Z.M., A.M.Jr, R.M., H.M., O.M., J.M., P.M., V.N., M.R.O., C.S., J.L.W., S.W.M., and S.A.W. provided materials; M.P. extracted DNA, prepared samples for sequencing, and performed qPCR analyses; M.C.P. and J.G. prepared pooled samples from Mozambique and extracted DNA; M.P. led and performed the bioinformatics analyses; M.P. and S.R.D. analysed and interpreted the results, with input from contributions from A.W., C.C., and D.T.J.L.; M.P. and S.R.D. drafted the original manuscript, with contributions from A.W., C.C., and D.T.J.L. All authors contributed to the revision and approved of the final manuscript.

## Competing interests

The authors declare no competing interests

## Additional information

**Marina Papaiakovou** [1,2,3] ✉, **Andrea Waeschenbach** [2], **Olumide Ajibola**[4], **Sitara SR Ajjampur** [5,6], **Roy M. Anderson**[7], **Robin Bailey** [6,8], **Jade Benjamin-Chung** [9,10], **Maria Cambra-Pellejà**[11,12,13], **Nicolas R. Caro**[14], **David Chaima**[6,15], **Rubén O. Cimino**[14], **Piet Cools**[16], **Anélsio Cossa**[17], **Julia Dunn**[7], **Sean Galagan**[6,18], **Javier Gandasegui** [3,11], **Berta Grau-Pujol** [11,17,19], **Emma L. Houlder**[20], **Moudachirou Ibikounlé** [6,21,22], **Timothy P. Jenkins** [23], **Khumbo Kalua**[6,24], **Eyrun F. Kjetland**[25,26], **Alejandro J. Krolewiecki** [14,27], **Bruno Levecke** [16], **Adrian JF Luty** [6,28], **Andrew S. MacDonald** [29], **Inácio Mandomando**[11,17,30,31], **Malathi Manuel**[5,6], **Maria Martínez-Valladares**[32], **Rojelio Mejia**[33], **Zeleke Mekonnen** [34], **Augusto Messa Jr.**[11,13,17], **Harriet Mpairwe**[35], **Osvaldo Muchisse**[17], **Jose Muñoz**[11,36], **Pauline Mwinzi**[37,38], **Valdemiro Novela** [17], **Maurice R. Odiere** [37], **Charfudin Sacoor**[17], **Judd L. Walson** [6,39], **Steven A. Williams**[40], **Stefan Witek-McManus**[6,41], **D. Timothy J. Littlewood** [2,42], **Cinzia Cantacessi** [1,42] & **Stephen R. Doyle** [3] ✉

[1]Department of Veterinary Medicine, University of Cambridge, Cambridge, UK. [2]Natural History Museum, Cromwell Road, London, UK. [3]Wellcome Sanger Institute, Hinxton, Cambridgeshire, UK. [4]Department of Biochemistry, Nigerian Institute of Medical Research, Yaba, Lagos, Nigeria. [5]The Wellcome Trust Research Laboratory, Division of Gastrointestinal Sciences, Christian Medical College Vellore, Vellore, Tamil Nadu, India. [6]The DeWorm3 Project, University of Washington, Seattle, WA, USA. [7]Department of Infectious Disease Epidemiology, School of Public Health, Faculty of Medicine, Imperial College London, White City Campus, London, UK. [8]Clinical Research Department, Faculty of Infectious and Tropical Disease, London School of Hygiene and Tropical Medicine, London, UK. [9]Department of Epidemiology and Population Health, Stanford University, Stanford, CA, USA. [10]Chan Zuckerberg Biohub, San Francisco, CA, USA. [11]ISGlobal, Barcelona, Spain. [12]GraphenicaLab SL, Barcelona, Spain. [13]Facultat de Medicina i Ciències de la Salut, Universitat de Barcelona, Barcelona, Spain. [14]Instituto de Investigaciones de Enfermedades Tropicales (IIET-CONICET), Facultad Regional Orán, Universidad Nacional de Salta, Orán, Argentina. [15]Department of Pathology, School of Medicine and Oral Health, Kamuzu University of Health Sciences, Blantyre, Malawi. [16]Department of Translational Physiology, Infectiology and Public Health, Ghent University, Merelbeke, Belgium. [17]Centro de Investigação em Saúde de Manhiça (CISM), Maputo, Mozambique. [18]Department of Global Health, University of Washington, Seattle, WA, USA. [19]Mundo Sano Foundation, Buenos Aires, Argentina. [20]Leiden University Center for Infectious Diseases, Leiden University Medical Center, Leiden, Netherlands. [21]Centre de Recherche pour la lutte contre les Maladies Infectieuses Tropicales (CReMIT/TIDRC), Université d'Abomey-Calavi, Cotonou, Benin. [22]Institut de Recherche Clinique du Bénin, Abomey-Calavi, Benin. [23]Department of Biotechnology and Biomedicine, Technical University of Denmark, Kongens Lyngby, Denmark. [24]Blantyre Institute for Community Outreach, Lions Sight First Eye Hospital, Blantyre, Malawi. [25]Norwegian Centre for Imported and Tropical Diseases, Department of Infectious Diseases Ullevaal/Department of Global Health, Oslo University Hospital, Oslo, Norway. [26]Discipline of Public Health Medicine, Nelson R Mandela School of Medicine, College of Health Sciences, University of KwaZulu-Natal, Durban, South Africa. [27]Innovation Area, Mundo Sano Foundation, Buenos Aires, Argentina. [28]Université Paris Cité, IRD, MERIT, Paris, France. [29]Institute of Immunology and Infection Research, School of Biological Sciences, University of Edinburgh, Edinburgh, United Kingdom. [30]Global Health and Tropical Medicine, GHTM, Associate Laboratory in Translation and Innovation Towards Global Health, LA-REAL, Instituto de Higiene e Medicina Tropical, IHMT, Universidade NOVA de Lisboa, UNL, Lisboa, Portugal. [31]Instituto Nacional de Saúde (INS), Marracuene, Maputo, Mozambique. [32]Instituto de Ganadería de Montaña, CSIC-Universidad de León, Grulleros, León, Spain. [33]Departments of Pediatrics and Medicine, National School of Tropical Medicine, Baylor College of Medicine, Houston, TX, USA. [34]School of Medical Laboratory Sciences, Institute of Health, Jimma University, Jimma, Ethiopia. [35]MRC/UVRI and LSHTM Uganda Research Unit, Entebbe, Uganda. [36]International Health Department, Hospital Clinic de Barcelona, Barcelona, Spain. [37]Centre for Global Health Research, Kenya Medical Research Institute, Kisumu, Kenya. [38]Expanded Special Project for Elimination of NTDs, WHO Regional Office for Africa, Brazzaville, Republic of Congo. [39]Departments of International Health, Medicine and Pediatrics, Bloomberg School of Public Health, Johns Hopkins University, Baltimore, MD, USA. [40]Department of Biological Sciences, Smith College, Northampton, MA, USA. [41]Department of Disease Control, Faculty of Infectious and Tropical Diseases, London School of Hygiene and Tropical Medicine, London, UK. [42]These authors contributed equally: D. Timothy J. Littlewood, Cinzia Cantacessi. ✉e-mail: mpapaiakovou@gmail.com; stephen.doyle@sanger.ac.uk

