## [Peer Review file · Nature Communications]

Global diversity of soil-transmitted helminths reveals population-biased genetic variation that impacts diagnostic targets

Corresponding Author: Ms Marina Papaïakovou

Version 0:

Reviewer comments:

Reviewer #1

(Remarks to the Author)

The article by Marina Papaïakovou uses publicly available and new sequence data to explore genetic diversity in key helminth species that cause important diseases in humans around the globe. Using a range of DNA from pooled eggs/larvae in faeces and purified eggs or individual worms, the authors explore the use of low coverage (DNA skimming) datasets for exploring genetic and population diversity in species (with sufficient data). The authors then go on to look at whether existing diagnostic primers are suitable with a better understanding of likely genetic variability. The methods are well described and allow for reproducibility of their datasets for future analyses.

The main strength/impact of this manuscript lies in the curation and analysis of genomic data to improve diagnostic tests for soil-transmitted helminths. By using whole genome sequencing, traditional primer annealing sites (which are usually not sequenced) were able to be characterised herein. This is a fantastic resource for improving the sensitivity and specificity of current diagnostic tests, particularly when relying on molecular methods (PCR) in low infection level settings.

With the low coverage read data, the authors also used a mix of mitochondrial and nuclear data derived to explore the genetic and population structures of some species, for which sufficient data was collated. Here, the manuscript relies on a mix of individual and pooled data from a range of studies, and this limited the conclusions that could be drawn from the data. At times, it is unclear with such low coverage, whether some of the conclusions on genetic diversity within species are sufficiently robust without additional analysis to support their findings. The article raises key hypotheses on this topic that need to be addressed in future work.

The exploration of genetic diversity within primer annealing sites was conducted well and shows clear evidence for mutations within primer annealing sites that had a profound impact on the predicted sensitivity of the qPCR. This is a significant finding and strengthens support for the further characterisation of more genomic data from taxa within and between populations globally. I would have liked to see a more detailed output (supplementary table with primers with new ambiguity (IUPAC) coding or VCF files of mapped data within primer annealing sites) as a resource for the next generation of primer designers to make it easy for them to avoid/account for regions of mutation. Given the wealth of taxonomically/phylogenetically informative data in loci currently being amplified in tests, I can imagine that the desire to retain primers for these well characterised regions would be preferred over new sites without existing metadata.

I have some minor comments which are outlined below.

1. Line 102. Should be substantial
2. Line 94. Just say parasites, or pathogenic parasites so next sentence is clear to a broader audience
3. Line 99. Unclear what is meant by presence. Presence of parasite DNA in ... would make it clear what you did.
4. Line 103. "Target regions?"
5. The PCR-positive 842 faecal samples were sequenced and only 175 contained parasite DNA? Was this just depth of sequencing? Is this something that should be discussed?
6. Line 180. Please provide a citation for Ascaris egg laying
7. Line 187. Not clear what was meant by helminth positive. Should be clearer to state DNA was successfully sequenced from 154 samples.
8. Figure 1. Suggest removing faecal and parasite symbols and change "n of samples" To Number of faecal samples or Number of helminth samples

9. Fig 2. Genetic diversity in *Ascaris* spp. and *Trichuris trichiura*. These results are quite comprehensive in breaking down the differences in nuclear diversity among these taxa. Conclusions in this section of the paper are appropriate and in line with other studies, however, the sequence coverage used to make conclusions was low and I think it should be clearer that these findings are more preliminary and point to a trend (within the results section and not just in discussion).

10. Line 345. Given the fragmented nature of repeat elements in genomes, would a fragment-aware homology aligner like RepeatMasker be appropriate/sensitive for detecting the presence of repeat elements within genomes?

11. Line 396. Please change similarity to nucleotide identity throughout to avoid ambiguity.

12. Line 596 --filter-sample-min-coverage 2 -- for determining the genetic diversity in pools of samples, the coverage minimum was set to 2. Was this depth of coverage suitable for accurately determining diversity? It would be good to understand better how the rate of missingness led to such low levels of coverage used for filtering.

13. Line 619. Would it be possible to report the variable sites within the primers used for diagnostics tested here (see Supplementary Table 4?). It would make it easy for diagnostic labs to re-design tests based on the results of this work.

14. Herein, the authors refer to rRNA markers and microsatellite markers all as repetitive sequences. As rRNA is under highly conserved, I wonder if it is appropriate to analyse rRNA and microsatellites at once? It is my understanding that rates of mutation would vary significantly. Would that have impacted the interpretation of the results?

(Remarks on code availability)

Code provided in full with some commenting to help with narrative. The examples of output are included in a notebook format which is really useful.

Reviewer #2

(Remarks to the Author)

The article 'Global diversity of soil-transmitted helminths reveals population-biased genetic variation that impacts diagnostic targets' have investigated the major transmitted Platyhelminthes DNA diversity from various countries and develop the molecular diagnostics. The high sensitivity and specificity of each helminthes are essential and will be a great basic knowledge for further infection control research. The experimental plan, sample collections, methodology, data analysis and statistic used in this study are sufficient, and the outcome and data are noteworthy to the field. The authors interpreted the genomic data, and be able to develop the diagnostic for the interested parasites. The data interpretation and conclusion are clear. In my opinion, this work is informative for broad knowledge for the field. The manuscript was written clearly. I have only minor comments to the authors to discuss how the genomic diagnosis could be useful in the case of patients have been treated with the appropriate drugs, and the remaining genomic material(s) may give the false positive. Other than that, the manuscript would be accepted for publication.

(Remarks on code availability)

Reviewer #3

(Remarks to the Author)

This is a very extensive, interesting and potentially important piece of work since there is a dearth of information on genetic variation on human STH species and how this relates (and/or may confound) current and future molecular diagnostic assays. There is an increasing need to deploy such diagnostic assays to monitor current STH control programs and understanding and maximizing their sensitivity and specificity is critical particularly as regional STH elimination becomes the goal.

The approach is novel, in terms of undertaking low coverage shotgun sequencing to assess regional variation in sequence polymorphism and copy number variations in several STH species and their potential to affect current molecular diagnostic assays. Overall, the scale and originality of the work is impressive and it appears technically sound and the conclusions broadly valid. It provides a lot of valuable information on both copy number variation and primer site polymorphisms and their potential impact on current molecular diagnostic tools (although this could have been more clearly discussed as per my comments below). It also provides a lot of interesting information on the utility of the strategy of metagenomic shotgun sequencing on fecal stool samples to examine STH and their genetics. This is an innovative approach that others in the field can use and the information in this paper should be very useful to help guide those approaches.

The biggest weakness of the paper is that it is somewhat disjointed and a little unstructured, and even confusing in places. Also there was a lack of overall synthesis – particularly in the discussion section. I get the impression that it was put together by multiple contributions from several key groups and more consideration should be to harmonisation the different sections and conclusions in the discussion. The discussion section also seemed uneven with respect to the depth of discussion to the different objectives; co-infection prevalence, mitochondrial sequence variation (and what it says about population structure), repeat copy number variation and primer site sequence polymorphism. I also struggled in places to relate specific pieces from materials and methods section with the relevant results sections (an example is given below)

Overall, I think this paper is worthy of consideration for publication in Nature Communications but the text, particularly the discussion, requires some more work to address the general issues outlined above.

Major specific comments below :

1. This is clearly a complex set of samples set, presumably put together through opportunistic collaborations to provide a

large dataset. This approach is fine (and actually somewhat inevitable to produce such a large and geographically diverse sample set) but the description of the numbers of samples used in preparing the libraries and how this relates to the data presented is quite confusing. It would be helpful if there were a few sentences at the beginning of the results section clarifying this issue and summarizing the overall strategy of the work in relation to the samples (and maybe a simple table or schematic in supplementary materials).

One specific example is on lines 171-172 in the results it states "1000 DNA samples extracted from individual worms (n = 128) as well as faecal (n = 842) and egg samples (n = 30), from 27 countries across all continents but in the materials and methods on lines 514 -515 it states "six different extraction kits were used to isolate DNA from the 150 faecal or worm samples (Supplementary Table 7) that were sequenced as part of this study...". I couldn't relate these two things together or find additional information to help.

2. The competitive mapping of the *Ascaris* mitochondrial sequence data was interesting although I found the interpretation a little confusing (and there was minimal wider discussion of this result). My understanding of the analysis presented, suggests that most samples preferentially mapped to the *A. suum* reference sequence than either the Korean or Tanzanian *A. lumbricoides* reference sequences. However the only discussion of this is a statement on lines 452-453 that : "Although all samples analysed here were obtained from humans, there was an overwhelming bias towards *A. suum*-derived genetic diversity, leading us to question the evolutionary background of the original reference sample" I don't understand this statement, given that there was preferential mapping to *A. suum* reference sequence not only when the Korean reference sequence was used (presumably what is being referred to as the "original reference sequence") but also when the Tanzanian *A. lumbricoides* reference sequence was used. Clarification of this is required. Also more discussion of what this means in relation to our current thinking of the species status of *A. suum* and *A. lumbricoides*.

3. Regarding the mitochondrial sequence diversity analysis and the apparent differences in the population diversity and structure between *A. lumbricoides* and *T. trichiura*. On line 266-267 it states "in both worm and egg samples, in almost all cases, nucleotide diversity and variant frequency were higher in *T. trichiura* than in *A. lumbricoides* populations". Whilst I can see this is broadly true, the complexity of the data set makes it difficult to be definitive. For example in figure 2b (eggs data)—there are many more data points for *A. lumbricoides* (with a lot of variance) compared to *T. trichiura* for which there very few data points. Some comment on the differences in the number of datapoints for each species would be helpful and appropriate caveats made to the conclusions.

4. lines 308 to 314 – a sentence or two explaining how these repeats were chosen and found would be helpful here.

5. Discussion section : I think the discussion section needs substantially more work. The central concept of the paper is to see how sequence and copy number variation of current diagnostic STH markers could confound their reliability. A much clearer discussion section on this is needed, including a concise summary of the relevant diagnostic markers and their use (with references), how the specific results relate to the issue of their reliability and what are the practical implications of this. This would make the paper more accessible and interesting to a wider set of readers

Minor comments

Line 102 – "substantial" incorrectly spelt.

Line 102 – "variants" should be "variation"

Line 118 – *A. ceylanicum* should be mentioned – likely more prevalent than *A. duodenale*

Lines 157 and 171 : I think "worm" should be "adult worm" for clarity

(Remarks on code availability)

NA

Version 1:

Reviewer comments:

Reviewer #1

(Remarks to the Author)

I thank the authors for comprehensively addressing my comments. I am satisfied with the rejoinder and the R1 version of the manuscript. The manuscript raises an important issue with traditional PCR diagnostic methods and has provided an accessible resource to improve PCR for future, universal diagnostic tests for several helminth species. It is clear that further exploration of natural genetic diversity within helminth species is needed.

(Remarks on code availability)

The code is data—and system-dependent but easy to follow for reproducibility. Commenting on the code is sufficient to aid tracking through the steps.

Reviewer #3

(Remarks to the Author)

Although I would have liked to have seen more clarity regarding certain aspects of the discussion the authors have modified

the manuscript sufficiently for it to be acceptable for publication in my opinion. It is an important and innovative piece of work.

(Remarks on code availability)

REVIEWER COMMENTS

Reviewer #1:

The article by Marina Papaïakovou uses publicly available and new sequence data to explore genetic diversity in key helminth species that cause important diseases in humans around the globe. Using a range of DNA from pooled eggs/larvae in faeces and purified eggs or individual worms, the authors explore the use of low coverage (DNA skimming) datasets for exploring genetic and population diversity in species (with sufficient data). The authors then go on to look at whether existing diagnostic primers are suitable with a better understanding of likely genetic variability. The methods are well described and allow for reproducibility of their datasets for future analyses.

The main strength/impact of this manuscript lies in the curation and analysis of genomic data to improve diagnostic tests for soil-transmitted helminths. By using whole genome sequencing, traditional primer annealing sites (which are usually not sequenced) were able to be characterised herein. This is a fantastic resource for improving the sensitivity and specificity of current diagnostic tests, particularly when relying on molecular methods (PCR) in low infection level settings.

RESPONSE:

We would like to thank the reviewer for their highly supportive and positive comments on our manuscript.

With the low coverage read data, the authors also used a mix of mitochondrial and nuclear data derived to explore the genetic and population structures of some species, for which sufficient data was collated. Here, the manuscript relies on a mix of individual and pooled data from a range of studies, and this limited the conclusions that could be drawn from the data. At times, it is unclear with such low coverage, whether the some of the conclusions on genetic diversity within species are sufficiently robust without additional analyse to support their findings. The article raises key hypotheses on this topic that need to be addressed in future work.

RESPONSE:

We agree with the reviewer that our use of both individual worm and pooled faecal/egg data, generated both by us and elsewhere, is not an ideal scenario and has limited the scope of the work. However, we are confident that the reviewer appreciates the difficulty in collecting these samples in the first place, which necessitates compromises on the types of analyses that would be ideal to perform to more robustly test some of the

proposed hypotheses. We are also confident that our study, as the reviewer recognises, raises hypotheses to be tested in future work, and will encourage more thorough validation of current and future helminth diagnostics as more genomic datasets become available.

The exploration of genetic diversity within primer annealing sites was conducted well and shows clear evidence for mutations within primer annealing sites that had a profound impact on the predicted sensitivity of the qPCR. This is a significant finding and strengthens support for the further characterisation of more genomic data from taxa within and between populations globally. I would have liked to see a more detailed output (supplementary table with primers with new ambiguity (IUPAC) coding or VCF files of mapped data within primer annealing sites) as a resource for the next generation of primer designers to make it easy for them to avoid/account for regions of mutation. Given the wealth of taxonomically/phylogenetically informative data in loci currently being amplified in tests, I can imagine that the desire to retain primers for these well characterised regions would be preferred over new sites without existing metadata.

RESPONSE:

We appreciate the recognition of the importance of genetic variation on qPCR priming.

In response to the reviewer's suggestion, we have now provided a table with the primer and probe binding sites across the entire genomes of all three species (*Ascaris lumbricoides*, *Trichuris trichiura* and *Necator americanus*) (Supplementary Tables 8 and 9) and the VCF files (https://github.com/MarinaSci/Global_skim_analysis/tree/main/01_VCF_DATA) that contain SNP information per repeat to facilitate future diagnostics validation. Together, these files could be used to refine existing or develop new primers accounting for known genetic variation.

I have some minor comments which are outlined below.

1. Line 102. Should be substantial

RESPONSE:

We have now fixed the typo.

2. Line 94. Just say parasites, or pathogenic parasites so next sentence is clear to a broader audience

RESPONSE:

We have now fixed the sentence.

3. Line 99. Unclear what is meant by presence. Presence of parasite DNA in ... would make it clear what you did.

RESPONSE:

We agree that “presence” was unclear.

In response to this comment, we have removed the word from the sentence and changed it to “genetics”. The sentence now reads:

“Using low-coverage genome sequencing, we assessed the genetics of STHs within worm, faecal and purified egg samples from 27 countries, identifying differences in the genetic connectivity and diversity of STH-positive samples across regions and cryptic diversity between closely related human- and pig-infective species.”

4. Line 103. "Target regions?"

RESPONSE:

We have now updated the sentence.

5. The PCR-positive 842 faecal samples were sequenced and only 175 contained parasite DNA? Was this just depth of sequencing? Is this something that should be discussed?

RESPONSE:

This is a sensible comment from the reviewer, and it reflects that our description of the samples used—a mix of publicly available and new data from worms, faecal DNA, and mixed eggs—was not as clear as we would have liked it to be.

A key piece of information that was missing from the text, but is now included, is that of the 842 faecal samples/datasets, all were suspected to contain STHs, but only a relatively small proportion were actually independently validated as having STH

infections. Most of these samples were chosen and sequenced based on the fact that at least one STH had been detected previously. Therefore, is it possible that some of those samples suspected of being parasite-positive were in fact negative.

A further complication arises from the sensitivity and specificity of using sequencing to detect infections. Previous work from our lab has demonstrated that the percentage of parasite reads obtained from faecal samples is very small (<0.01%). Moreover, our thorough QC of the data revealed that a very small proportion of reads mapped incorrectly for some species, leading to false positive results. Therefore, we consciously chose a conservative threshold of 10 helminth reads per million reads mapped as defining a positive result. This relatively conservative threshold helps ensure confidence in the positives while acknowledging that we may have missed some low-abundance infections.

In response to this comment (and other comments further below), we have revised the description of the samples at the beginning of the results to enhance clarity.

6. Line 180. Please provide a citation for *Ascaris* egg laying

RESPONSE:

We have now added a relevant reference for *Ascaris* egg laying.

7. Line 187. Not clear what was meant by helminth positive. Should be clearer to state DNA was successfully sequenced from 154 samples.

RESPONSE:

Helminth positive reflects the number of samples in which we were confident of detecting at least one helminth by sequencing, that passed our threshold of 10 helminth reads per million reads mapped, as described above. We have specifically included the threshold in parentheses to emphasise that.

In response to this comment, we have clarified the sentence to state that they were “positive by sequencing” and acknowledged the issue of false negatives.

“Out of 842 faecal samples, 175 were positive by sequencing for at least one helminth species (minimum threshold of 10 helminth reads / million reads mapped; Supplementary Fig. 1a) spanning 14 different countries (Fig. 1a, c and Supplementary Fig. 2a). We suspect that our approach resulted in false-negatives, due to the low

parasite prevalence of many infections and the conservative minimum threshold necessary to ensure sensitivity and specificity of detection.”

And

“Among the 158 samples derived from worms or pools of eggs, 154 were helminth-positive **by sequencing**, spanning 15 different countries”.

8. Figure 1. Suggest removing faecal and parasite symbols and change "n of samples" To Number of faecal samples or Number of helminth samples

RESPONSE:

We have chosen to keep the symbols, as they provide a point of reference within and between figures to differentiate the different datasets.

In response to this comment, we have changed the ‘n of samples’ to ‘number of samples’.

9. Fig 2. Genetic diversity in *Ascaris* spp. and *Trichuris trichiura*. These results are quite comprehensive in breaking down the differences in nuclear diversity among these taxa. Conclusions in this section of the paper are appropriate and in line with other studies, however, the sequence coverage used to make conclusions was low and I think it should be clearer that these findings are more preliminary and point to a trend (within the results section and not just in discussion).

RESPONSE:

This is a sensible comment from the reviewer.

In response to this comment, we have elaborated on one of the conclusions in the results section to acknowledge the low sequencing coverage and the necessity of validating with higher coverage data.

“This pattern of variation in *A. lumbricoides* is consistent with that of Easton *et al.*,(2020), where very few variants were found to differentiate haplotypes from reference genomes sampled from around the world, even though the data presented here exhibit lower coverage and more sampling with higher depth would be necessary to validate such trends for both species. “

10. Line 345. Given the fragmented nature of repeat elements in genomes, would a fragment-aware homology aligner like RepeatMasker be appropriate/sensitive for detecting the presence of repeat elements within genomes?

RESPONSE:

This is an interesting suggestion from the reviewer.

If we were seeking to identify new candidate repeats for designing primers, then yes, a tool like RepeatModeller and RepeatMasker would likely be appropriate. However, we aimed to find the location of very specific repeats in the genome that have been previously used as diagnostic primer sites. Therefore, we chose to use a targeted detection approach.

Potentially, we could have trained RepeatModeller with the diagnostic repeats alone, after which RepeatMasker could have been used to detect them. However, there is greater uncertainty in the sensitivity and specificity of that approach, which may need validation, relative to the more straightforward and arguably more interpretable approach we have used.

11. Line 396. Please change similarity to nucleotide identity throughout to avoid ambiguity.

RESPONSE:

We suspect the reviewer refers to the text addressing the similarity concerning the repeats and not throughout the manuscript (i.e., population genetics analysis). We have now updated the terminology in that section, referring to the repeat similarity as 'nucleotide identity'.

12. Line 596 --filter-sample-min-coverage 2 -- for determining the genetic diversity in pools of samples, the coverage minimum was set to 2. Was this depth of coverage suitable for accurately determining diversity? It would be good to understand better how the rate of missingness led to such low levels of coverage used for filtering.

RESPONSE:

This is a good observation and a sensible question to ask.

To address this comment, we reevaluated various thresholds of sample coverage (1, 5, 10, 20) and assessed nucleotide diversity (as relative theta pi), mean depth in relation to theta pi and Dxy, which are plotted below. For the calculations of nucleotide diversity and Dxy, we utilised the pooled sequencing population genomics tool Gredalf with the original alignment BAM files. Setting the threshold as low as 1 or 2 did not affect the relevant theta pi (nucleotide diversity) used in our work, and we also found no association between mean depth and nucleotide diversity. The code used to generate the below plots can be found here: https://github.com/MarinaSci/Global_skim_analysis/blob/main/poo_worm_STH_global_s_kim_08_Pi_DXY_COVERAGE_THRESHOLDS_NATCOMMS_REVIEWS.md.

Figure. Testing various coverage filters when calculating theta pi diversity using Gredalf.

In addition, to show that there is no association between coverage and the calculated relative theta pi, we plotted the mean depth (calculated by samtools depth, version 1.6, using BAM files) for all pooled samples (n=16) positive for *Ascaris suum*. No association was found, and in combination with the coverage thresholds plot, we are confident in the selected values when using Gredalf for pi and Dxy.

Figure. Relationship between mean depth (logged x axis) and theta pi (relative) for 16 pooled samples positive for *Ascaris suum*.

For Dxy (between pi) with Grededalf, the number of positions and the Dxy values also did not change with different filters on coverage. Up to setting the coverage to 20, there is no difference in the Dxy values. When going to 40 and above, Dxy is slightly reduced (the number of SNPs does not change, but only when coverage is set to 80). Only when a coverage of 80 was tested did we lose 3 SNPs (from 308 it became 305).

Figure. Testing various coverage filters when calculating Dxy using Grenedalf.

Therefore, despite the low coverage, our data are robust to low-coverage samples. We acknowledge that there are variable rates of missingness, as expected in low coverage datasets. For the calculations of nucleotide diversity and Dxy, we used Grenedalf with the original alignment BAM files. As input to Grenedalf, we filtered the BAM files by quality, CIGAR patterns to minimise mapping errors, and read length. Additionally, Grenedalf applies other filters (on mapping quality and base quality), ensuring that the results remain reliable. One option to better account for missingness would have been to use VCFs as input. However, we refrained from using the VCFs (that had already been filtered for missingness) because VCFs already have statistical assumptions (regarding genotypes) built in that could interfere with Grenedalf.

13. Line 619. Would it be possible to report the variable sites within the primers used for diagnostics tested here (see Supplementary Table 4?). It would make it easy for diagnostic labs to re-design tests based on the results of this work.

RESPONSE:

As described above, we have provided a table with the primer and probe binding sites across the entire genomes of all three species (*Ascaris lumbricoides*, *Trichuris trichiura*, and *Necator americanus*) (Supplementary Tables 8 and 9), along with the VCF files (https://github.com/MarinaSci/Global_skim_analysis/tree/main/01_VCF_DATA) that contain SNP information per repeat to facilitate the design and validation of future diagnostics.

14. Herein, the authors refer to rRNA markers and microsatellite markers all as repetitive sequences. As rRNA is highly conserved, I wonder if it is appropriate to analyse rRNA and microsatellites at once? It is my understanding that rates of mutation would vary significantly. Would that have impacted the interpretation of the results?

RESPONSE:

We agree that rDNA regions are highly conserved and that mutation rates are likely different between ribosomal sequences and microsatellites. In our study, the locations of the different repeat arrays in the genome were detected at the same time; however, genetic variation within each repeat assay was subsequently analysed separately. Therefore, there is no impact on the interpretation of the results.

Reviewer #1 (Remarks on code availability): Code provided in full with some commenting to help with narrative. The examples of output are included in a notebook format which is really useful.

RESPONSE:

We appreciate the reviewer's recognition of the availability of the code.

Reviewer #2 (Remarks to the Author):

The article 'Global diversity of soil-transmitted helminths reveals population-biased genetic variation that impacts diagnostic targets' have investigated the major transmitted Platyhelminthes DNA diversity from various countries and develop the molecular diagnostics. The high sensitivity and specificity of each helminthes are essential and will be a great basic knowledge for further infection control research. The experimental plan, sample collections, methodology, data analysis and statistic used in this study are sufficient, and the outcome and data are noteworthy to the field. The authors interpreted the genomic data, and be able to develop the diagnostic for the interested parasites. The data interpretation and conclusion are clear. In my opinion, this work is informative for broad knowledge for the field. The manuscript was written clearly. I have only minor comments to the authors to discuss how the genomic diagnosis could be useful in the case of patients have been treated with the appropriate drugs, and the remaining genomic material(s) may give the false positive. Other than that, the manuscript would be accepted for publication.

RESPONSE:

We thank the reviewer for their very positive appraisal of our manuscript.

Regarding the reviewers' query about whether genetic or genomic diagnosis could be useful for detecting parasites, even after appropriate treatment, and whether this indicates a false positive, we believe there are two possible but distinct scenarios here:

1. the parasite has responded to treatment, and they are all dead, but the dead parasites have not been fully cleared from the host; or
2. The patient has been treated, but infection persists, potentially reflecting a proportion of the parasite population that is not responding to treatment.

The first scenario is a true false positive if a positive is inferred as an active (i.e., live) parasite infection. However, from a genetics perspective, it may not matter, as it could still provide useful information about the genetics of a drug-responsive parasite population. Furthermore, complementary diagnostics (RNA-based) may enhance the diagnosis and determine whether this is an active or past infection.

In the second scenario, genetics may be able to differentiate whether this persistent infection is a reinfection or whether drug resistance has occurred.

While these are interesting and potentially valuable ideas that may contribute to the discussion of using genetic diagnostics more broadly, they are outside of the scope of the aims and findings of this study.

Reviewer #3 (Remarks to the Author):

This is a very extensive , interesting and potentially important piece of work since there is a dearth of information on genetic variation on human STH species and how this relates (and/or may confound) current and future molecular diagnostic assays. There is an increasing need to deploy such diagnostic assays to monitor current STH control programs and understanding and maximizing their sensitivity and specificity is critical particularly as regional STH elimination becomes the goal.

The approach is novel, in terms of undertaking low coverage shotgun sequencing to assess regional variation in sequence polymorphism and copy number variations in several STH species and their potential to affect current molecular diagnostic assays. Overall, the scale and originality of the work is impressive and it appears technically sound and the conclusions broadly valid . It provides a lot of valuable information on both copy number variation and primer site polymorphisms and their potential impact on current molecular diagnostic tools (although this could have been more clearly discussed as per my comments below). It also provides a lot of interesting information on the utility of the strategy of metagenomic shotgun sequencing on fecal stool samples to examine STH and their genetics. This is an innovative approach that others in the field can use and the information in this paper should be very useful to help guide those approaches .

RESPONSE:

We thank the reviewer for the positive comments and for recognising the value of this manuscript.

The biggest weakness of the paper is that it is somewhat disjointed and a little unstructured, and even confusing in places. Also there was a lack of overall sthensis – particularly in the discussion section. I get the impression that is was put together by multiple contributions from several key groups and more consideration should be to harmonisation the different sections and conclusions in the discussion. The discussion section also seemed uneven with respect to the depth of discussion to the different objectives; co-infection prevalence, mitochondrial sequence variation (and what it says

about population structure), repeat copy number variation and primer site sequence polymorphism. I also struggled in places to relate specific pieces from materials and methods section with the relevant results sections (an example is given below)

Overall, I think this paper is worthy of consideration for publication in Nature Communications but the text, particularly the discussion, requires some more work to address the general issues outlined above.

RESPONSE:

We appreciate the constructive comments; however, we disagree with the overarching argument that the manuscript is unstructured or uneven regarding the objectives.

Our overarching aim was to understand if genetic variation in parasite populations has the potential to impact molecular diagnostics. Our Results section had a very logical structure to address this aim:

- Section 1. Detecting STH in faecal DNA by sequencing/analysis of a global cohort
- Section 2. Measuring genetic variation within and between populations
- Section 3. Finding diagnostic targets in the genomes
- Section 4. Detecting genetic variation in these diagnostic targets and validating its effect.

Perhaps what confuses the reviewer is that we specifically chose to discuss “minor or serendipitous results” in the Results section, which we believe helps the reader interpret the results in their context more easily. We feel this was particularly important, for example, to explain the mapping bias among *Ascaris* references in a logical order of events. It was an unexpected result, tangential to the main aim of the study, but worth describing to aid our understanding and characterisation of the genetic variation.

This strategy of discussing minor results in the Results section allowed us to focus our Discussion on the main results and their broader context, including strengths, limitations, and future directions. To reiterate the logical structure of our Discussion:

- Paragraph 1: overview of the problem and our approach to address it
- Paragraph 2 on population genetics:
 - “we have identified contrasting patterns of genetic variation and connectivity between globally distributed STH species. “
- Paragraph 3 on detecting repeats in the context of technical and biological variation:
 - “Our nuclear genome analyses revealed the position, number, and nucleotide identity within species of these repeats in the genomes of *A.*

lumbricoides, *N. americanus* and *T. trichiura*. However, our interpretation of these regions will change as genome assemblies improve, “

- Paragraph 4 on the impact of genetic variation of qPCR diagnostics:
 - “We demonstrate that, under *in vitro* controlled conditions, single nucleotide genetic variants can significantly influence the ability of qPCR to detect these diagnostic repeats. We acknowledge that our *in vitro* assays are extreme scenarios and assume that all targets in a genome would be identical and perform equally poorly in the presence of the variant.”
- Paragraph 5, on final summary and the importance of importance of sampling and more comprehensive evaluation of diagnostics:
 - “Our study emphasises the importance of sampling a wider range of human and nonhuman hosts globally Further work to define the genetic connectivity of STH populations may enable the definition of transmission zones and networks, providing a more precise means to prioritise control efforts. Our analyses provide a clear rationale for more comprehensive evaluation and validation of repetitive sequences currently being used as targets of molecular diagnostics. “

Therefore, we have opted not to rewrite the Discussion extensively. We have specifically addressed the comments below and updated the text where necessary.

Major specific comments below :

1. This is clearly a complex set of samples set, presumably put together through opportunistic collaborations to provide a large dataset. This approach is fine (and actually somewhat inevitable to produce such a large and geographically diverse sample set) but the description of the numbers of samples used in preparing the libraries and how this relates to the data presented is quite confusing. It would be helpful if there were a few sentences at the beginning of the results section clarifying this issue and summarizing the overall strategy of the work in relation to the samples (and maybe a simple table or schematic in supplementary materials).

One specific example is on lines 171-172 in the results it states “1000 DNA samples extracted from individual worms (n = 128) as well as faecal (n = 842) and egg samples (n = 30), from 27 countries across all continents but in the materials and methods on lines 514 -515 it states “six different extraction kits were used to isolate DNA from the 150 faecal or worm samples (Supplementary Table 7) that were sequenced as part

of this study...”. . I couldn’t relate these two things together or find additional information to help.

RESPONSE:

We recognise that this was confusing, especially given Reviewer 1 also commented on it.

To respond to this comment, we have revised the first paragraph of the results to clarify the description of the sample cohort.

We note that we have included a Supplementary Table that describes the samples and a Supplementary Text that details the specifics of individual country cohorts. The reference to the different extraction kits relates solely to the samples we sequenced for this study, namely 150 samples.

2. The competitive mapping of the *Ascaris* mitochondrial sequence data was interesting although I found the interpretation a little confusing (and there was minimal wider discussion of this result). My understanding of the analysis presented, suggests that most samples preferentially mapped to the *A. suum* reference sequence than either the Korean or Tanzanian *A. lumbricoides* reference sequences . However the only discussion of this is a statement on lines 452-453 that :

“Although all samples analysed here were obtained from humans, there was an overwhelming bias towards *A. suum*-derived genetic diversity, leading us to question the evolutionary background of the original reference sample”

I don’t understand this statement, given that there was preferential mapping to *A. suum* reference sequence not only when the Korean reference sequence was used (presumably what is being referred to as the “original reference sequence”) but also when the Tanzanian *A. lumbricoides* reference sequence was used . Clarification of this is required . Also more discussion of what this means in relation to our current thinking of the species status of *A. suum* and *A. lumbricoides*.

RESPONSE:

We understand the challenge of this topic, as it remains a debate in the literature; however, our analysis of competitive mapping was a means to an end, which was to understand how best to characterise genetic variation in these samples. We intentionally did not delve into the details of this discussion regarding the debate, as our sampling and analyses were not designed to examine whether *A. suum* and *A. lumbricoides* represent one or two species.

We believe that our results support the hypothesis that there may be problems with the reference sequences designated as “*A. suum*” and “*A. lumbricoides*” (as we state in the Discussion and as referred to above). Furthermore, our working hypothesis is that both *A. suum* and *A. lumbricoides* can infect humans, but they are genetically distinct due to reproductive isolation. This may in fact explain the preferential mapping, such that some populations were simply more *A. suum*-like. This hypothesis (zoonotic but reproductively isolated) aligns not only with our data but also with other genomic datasets in the literature. However, this is not the focus of our study, nor are our samples sufficient to properly test this hypothesis, and therefore, speculating about it is beyond the scope of this work.

3. Regarding the mitochondrial sequence diversity analysis and the apparent differences in the population diversity and structure between *A. lumbricoides* and *T. trichiura*. On line 266-267 it states “in both worm and egg samples, in almost all cases, nucleotide diversity and variant frequency were higher in *T. trichiura* than in *A. lumbricoides* populations”. Whilst I can see this is broadly true, the complexity of the data set makes it difficult to be definitive. For example in figure 2b (eggs data)– there are many more data points for *A. lumbricoides* (with a lot of variance) compared to *T. trichiura* for which there very few data points. Some comment on the differences in the number of datapoints for each species would be helpful and appropriate caveats made to the conclusions.

RESPONSE:

In response to this comment, we have added text acknowledging that these results suggest trends and that the smaller sample sizes of *Trichuris* may impact the observed difference in nucleotide diversity.

“This pattern of variation in *A. lumbricoides* is consistent with that of Easton *et al.*,³⁹, where very few variants were found to differentiate haplotypes from reference genomes sampled from around the world, even though the data presented here exhibit lower coverage and more sampling with higher depth would be necessary to validate such trends for both species.”

And

“In almost all cases, nucleotide diversity and variant frequency were higher in *T. trichiura* than in *A. lumbricoides* populations, though results for *Trichuris* were only supported by few positive samples.”

4. lines 308 to 314 – a sentence or two explaining how these repeats were chosen and found would be helpful here.

RESPONSE:

In response to this comment, we have added text explaining how these repeats were identified and the criteria used for similarity and coverage, before describing the results.

“Published diagnostic repeat targets (Supplementary Table 4) were used to identify the coordinates of all similar repeat variations in each genome using *nucmer*, retaining candidate repeats throughout the genomes with a least 90% coverage and nucleotide identity to the published target. “

Further details on the exact steps taken to obtain the coordinates of these repeats can be found in the Materials and Methods section (following the Results) and also under the caption of Figure 3.

5. Discussion section : I think the discussion section needs substantially more work. The central concept of the paper is to see how sequence and copy number variation of current diagnostic STH markers could confound their reliability. A much clearer discussion section on this is needed, including a concise summary of the relevant diagnostic markers and their use (with references), how the specific results relate to the issue of their reliability and what are the practical implications of this. This would make the paper more accessible and interesting to a wider set of readers

RESPONSE:

As previously stated, we disagree with the reviewer that the discussion requires substantially more work.

We remind the reviewer that we already provide a supplementary table of qPCR molecular diagnostics that are developed or in use.

The reviewer suggests that “central concept of the paper is to see how sequence and copy number variation of current diagnostic STH markers could confound their reliability.” We argue that our manuscript does not directly address this issue, but rather focuses on characterising whether variation exists in the first place and whether it may have an impact. Our validation of variants in controlled in vitro assays supports our underlying hypothesis that genetic variation within and between populations has an

effect. However, this is not the approach or the aim to fully test the reliability of these assays.

As we have described throughout, we have identified a portion of the genetic variation in a subset of the tested populations. Similarly, we demonstrate proof of principle that some of these identified genetic variants significantly impact qPCR. However, these analyses are not exhaustive or complete in assessing the influence of genetic variants on every repeat. Our sampling and sequencing were not sufficient to do so. We have intentionally not focused on differentiating whether one repeat or target is superior to another because we have not explicitly tested that. Therefore, we have highlighted the risks of not accounting for this variation and proposed recommendations for further testing the hypotheses generated here.

Given these challenges, a more in-depth discussion of the reliability and practical implications may be speculative and potentially unhelpful to the reader or the field.

Minor comments

Line 102 – “substantial” incorrectly spelt.

RESPONSE:

This has now been fixed.

Line 102 =-“variants” should be “variation”

RESPONSE:

This has now been fixed.

Line 118 – *A. ceylanicum* should be mentioned – likely more prevalent than *A. duodenale*

RESPONSE:

We thank the reviewer for the observation and we have now included *A. ceylanicum* in the manuscript.

Lines 157 and 171 : I t hink “worm” should be “adult worm” for clarity

RESPONSE:

We have now revised the text throughout to more clearly delineate the stage of the worm as suggested.

Reviewer #3 (Remarks on code availability):

NA